# Implication of TIGIT$^+$ human memory B cells in immune regulation

Md Mahmudul Hasan[1], Sumi Sukumaran Nair[2], Jacqueline G. O'Leary [3], LuAnn Thompson-Snipes[3], Verah Nyarige[4], Junwen Wang[4], Walter Park[5], Mark Stegall[5], Raymond Heilman[2], Goran B. Klintmalm[3], HyeMee Joo [1✉] & SangKon Oh [1✉]

Regulatory B cells (Bregs) contribute to immune regulation. However, the mechanisms of action of Bregs remain elusive. Here, we report that T cell immunoreceptor with Ig and ITIM domains (TIGIT) expressed on human memory B cells especially CD19$^+$CD24$^{hi}$ CD27$^+$CD39$^{hi}$IgD$^-$IgM$^+$CD1c$^+$ B cells is essential for effective immune regulation. Mechanistically, TIGIT on memory B cells controls immune response by directly acting on T cells and by arresting proinflammatory function of dendritic cells, resulting in the suppression of Th1, Th2, Th17, and CXCR5$^+$ICOS$^+$ T cell response while promoting immune regulatory function of T cells. TIGIT$^+$ memory B cells are also superior to other B cells at expressing additional inhibitory molecules, including IL-10, TGFβ1, granzyme B, PD-L1, CD39/ CD73, and TIM-1. Lack or decrease of TIGIT$^+$ memory B cells is associated with increased donor-specific antibody and TFH response, and decreased Treg response in renal and liver allograft patients. Therefore, TIGIT$^+$ human memory B cells play critical roles in immune regulation.

---

[1] Department of Immunology, Mayo Clinic, Scottsdale, AZ, USA. [2] Department of Medicine, Mayo Clinic, Phoenix, AZ, USA. [3] Baylor University Medical Center, Dallas, TX, USA. [4] Department of Health Sciences Research, Mayo Clinic, Scottsdale, AZ, USA. [5] William J. von Liebig Transplant Center, Mayo Clinic, Rochester, MN, USA. ✉email: Joo.HyeMee@mayo.edu; Oh.Sangkon@mayo.edu

The roles of regulatory B cells (Bregs) in immune regulation have been widely described in different disease models, including inflammatory disease, infectious disease, transplantation, and cancer[1–4]. However, it is still unknown whether Bregs represent a developmentally specified and stable lineage or subsets of differentiated B cells that can display immunosuppressive functions in certain circumstances. It is thus important to define phenotypes and functional characteristics of Bregs.

To date, a number of human Breg subsets have been reported. These include CD24^hiCD38^hi immature transitional B cells (TBs)[1], CD24^hiCD27^+ human equivalent of murine B10 cells (hereinafter referred to as B10-like B cells)[5], CD27^+CD38^hi plasmablast Bregs[6], CD25^hiCD71^hiCD73^lo Br1 cells[7], CD39^+CD73^+ Bregs[8], CD38^+CD1d^+IgM^+CD147^+ Bregs[9], TIM-1^+ Bregs[10], and induced Bregs[11]. Each of these Breg subsets seems to be capable of suppressing inflammatory responses by exerting shared, as well as distinct mechanisms of action. For example, the majority of Bregs are capable of expressing IL-10, a key anti-inflammatory cytokine, and it has been widely used for determining Bregs and Breg subsets. In vitro-induced Bregs[11] can also display immune regulatory functions by expressing TGFβ1 and indoleamine 2, 3-dioxygenase. PD-L1, CD80, CD86, and CD1d expressed by TBs are also known to contribute to immune regulation[1,12–14]. CD38^+CD1d^+IgM^+CD147^+ Bregs can display immune regulatory functions by expressing granzyme B[9]. In addition, CD39^+CD73^+ Breg-induced adenosine-mediated immune regulation has also been reported[8].

Due to the lack of Breg-specific lineage markers, however, it is also true that previously reported Breg subsets are not well separated from others not only for their surface phenotypes, but also for their functional characteristics. Therefore, it has been difficult to know which subsets of human Bregs are more important than others for immune regulation, a critical question that needs to be addressed for the advancement of Breg biology and potential clinical development of Breg-based therapeutic strategy. To address this critical question, however, it is essential to better define their surface phenotypes, as well as functional characteristics along with their mechanisms of action to suppress inflammatory responses. In this regard, we[15] previously reported that CD24^hiCD27^+ B10-like Bregs were more efficient than TBs at suppressing CD4^+ T cell proliferation, as well as IFNγ/IL-17 expression in vitro. This was further supported by the data generated with liver transplant patients with plasma cell hepatitis. Nonetheless, data from our study[15] and others[1,12–14] also indicate that commonly known inhibitory molecules (including IL-10 and surface PD-L1) expressed by Bregs do not always represent their immunosuppressive functions, suggesting that some Bregs could display additional immunosuppressive mechanisms of action that remain to be investigated. Furthermore, CD24^hiCD27^+ B10-like Bregs might not be a homogenous population. These prompted us to further investigate cellular and molecular phenotypes, as well as functional characteristics of human Bregs and Breg subsets.

In this study, we divided the entire CD19^+ human B cell population into six different populations by applying a flow cytometry gating strategy based on the data acquired from FACS array analysis of human B cells from peripheral blood and tonsils. We found that CD24^hiCD27^+CD39^hiIgD^−IgM^+CD1c^+ memory B cells, a fraction of CD24^hiCD27^+ B10-like B cells, display potent and unique functional characteristics at suppressing immune responses. They express not only IL-10, granzyme B, and TGFβ1, but also surface PD-L1, CD39/CD73, and TIM-1. More importantly, they also express a co-inhibitory receptor TIGIT (T cell immunoreceptor with Ig and ITIM domains) that can modify dendritic cell (DC) functions to suppress Th1, Th2, Th17, and CXCR5^+ICOS^+ T cell response, while promoting IL-10-producing T cell response. Ex vivo data generated with blood

samples of both liver and kidney allograft recipients further support the importance of TIGIT^+ human memory B cells in immune regulation.

## Results

**CD39 expression is higher on IL-10^+ B cells than IL-10^− B cells from both human tonsils and blood.** Due to the lack of Breg-specific markers, IL-10 expression has been widely used for defining Bregs in both humans and mice. Therefore, we performed a FACS array experiment examining 242 different cell surface molecules. In brief, tonsillar mononuclear cells (MNCs) and peripheral blood mononuclear cells (PBMCs) were stimulated with CpG-B for 48 h. 12-myristate 13-acetate (PMA)/ionomycin, brefeldin A, and monensin (PIBM) cocktails were added for the last 5 h of the culture. Cells were stained with antibodies specific for cell surface molecules and then stained for intracellular IL-10. As shown in Fig. 1a, similar percentages of B cells from tonsils and blood express IL-10 after 48 h stimulation with CpG-B. Next, we assessed the surface phenotypes of both IL-10^+ and IL-10^− B cells by performing a FACS array experiment examining 242 different cell surface molecules (Fig.1b). As shown in Fig. 1b, 21 surface molecules showed mean florescence intensity (ΔMFI) ≥ 500 between IL-10^+ and IL-10^− B cells. Of those, CD39 expression (red arrow) was ubiquitously and most significantly higher on IL-10^+ B cells than IL-10^− B cells (Fig.1b and Supplementary Fig. 1). Increased expression of CD39, an ecto-ATPase, on regulatory T cells (Tregs) and myeloid-derived suppressor cells and its critical roles in immune suppression has been previously reported[16,17]. CD11a, CD18, CD21, CD24, CD35, CD40, CD47, CD53, CD54, CD58, CD147, and integrin α4 expressions were also higher on IL-10^+ B cells than IL-10^− B cells. CD11a expression level (black arrow) was also higher in IL-10^+ B cells than IL-10^− B cells from the blood, but this was not the case for B cells from tonsils. In contrast, the expression level of CD72 (black arrow)[18] was higher on IL-10^+ B cells than IL-10^− B cells from tonsils, but this difference was not observed on B cells from the blood.

We next tested whether CD39 expression, in a steady state, is also higher on B cells expressing other surface markers that are known to be expressed on human Breg subsets. We assessed expression levels of individual human Breg surface markers on CD39^hi and CD39^lo/− B cells. Figure 1c shows that CD39^hi B cells expressed increased levels of surface CD27, CD71, CD73, and CD147, when compared to CD39^lo/− B cells. On the contrary, CD39^lo/− B cells expressed increased of IgD, IgM, and CD38. These data suggested that CD39 expressed on B cells could be used as a potential surface marker for subsets of Bregs.

**CD39 expression is higher on marginal zone-like and memory B cells than others.** We next divided peripheral blood B cells of healthy individuals into six different populations (Fig. 2a) based on the expression levels of surface molecules, including IgD, CD24, CD27, CD38, and CD39 (Fig. 1). CD19^+CD24^hiCD38^hiCD39^loIgD^+ TBs were named as population 1 (P1). IgD^+ B cells were divided into three different populations, CD19^+CD27^+CD39^hiIgD^+ (population 2, P2), CD19^+CD27^−CD39^hiIgD^+ (population 3, P3), and CD19^+CD27^−CD39^loIgD^+ (population 5, P5). IgD^− B cells were divided into CD19^+CD27^+CD39^hiIgD^− (population 4, P4) and CD19^+CD27^−CD39^+IgD^− (population 6, P6) B cells. An unsupervised t-distributed stochastic neighbor embedding (t-SNE) analysis (Fig. 2b) further showed clustering of individual B cell populations. P2 (blue) aligned with P4 (light green); whereas P1 (red) mostly aligned with P5 (army green), followed by P3 (pink). However, P6 showed two clusters: one was in between P1 and P4, and the other cluster was close to P3.

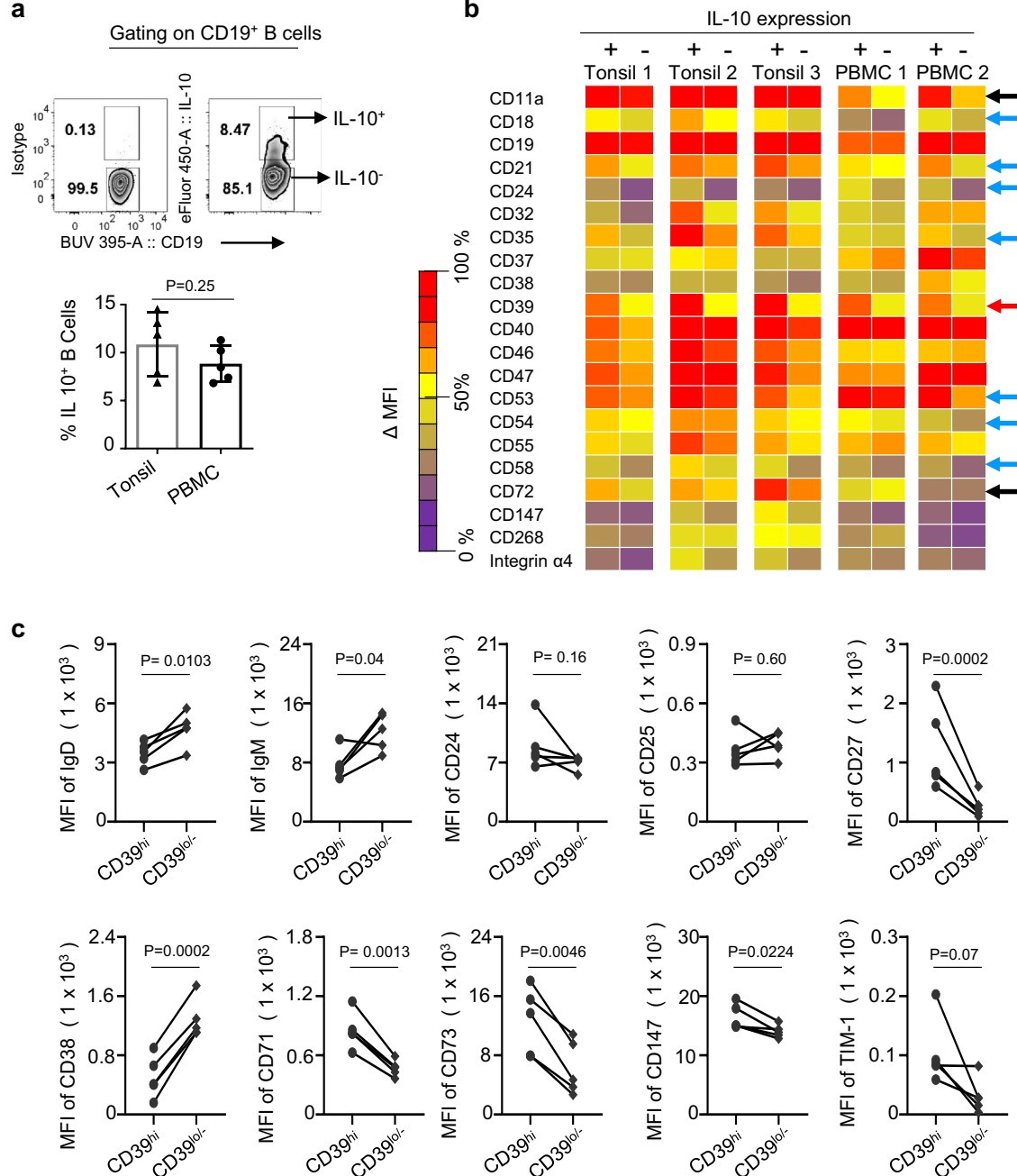

**Fig. 1 CD39 expression is higher on IL-10⁺ B cells than IL-10⁻ B cells from tonsils and peripheral blood mononuclear cells. a** A representative FACS plot showing IL-10 + and IL-10⁻ B cells in the blood of healthy subjects (upper panel). Cells were gated based on isotype antibody staining. Summarized data for the frequency of IL-10⁺ B cells (lower panel). Individual dots represent data generated with cells from different donors ($n = 5$). **b** Summary of FACS array data. Heatmap illustrating the expression levels of various cell surface markers on IL-10⁺ and IL-10⁻ B cells. Tonsillar mononuclear cells (MNCs) and peripheral blood mononuclear cells (PBMCs) were stimulated with CpG-B for 48 h. PMA/ionomycin, brefeldin A, and monensin cocktails were added for the last 5 h of the culture. Cells were stained with antibodies specific for cell surface molecules. Cells were then stained for intracellular IL-10 expression. Mean florescence intensity (MFI) was acquired upon subtracting isotype control. Cell surface markers with ΔMFI ≥ 500 between IL-10⁺ and IL-10⁻ B cells are presented. The colored scale shows the expression level (ΔMFI) of each cell surface markers on IL-10⁺ and IL-10⁻ B cells. Purple (ΔMFI > 25% by rank), yellow (ΔMFI = 25–65% by rank), and red (ΔMFI < 65% by rank) colors were used to indicate low to high level of expression. **c** MFI values of surface IgD, IgM, CD24, CD25, CD27, CD38, CD71, CD73, CD147, and TIM-1 on CD39^high and CD39^low/− B cells. Cells were stained without activation. Data generated with cells from different healthy subjects ($n = 5$) were combined. Error bars are mean ± SD in **a**. *P* values were determined with a two-tailed unpaired *t* test in **a** and a paired *t* test in **c**.

We further characterized surface phenotypes of the six B cell populations (P1–P6; Fig. 2c and Supplementary Fig. 2). As TBs (CD24^hiCD38^hi), P1 expressed increased levels of surface IgD and IgM, as well as CD5, CD9, and CD10[15]. However, P1 TBs expressed lower levels of surface CD39 and CD73 than P2, P3, P4,

and P6 B cells. P2 and P4 B cells were CD24^hiCD27⁺ B10-like Bregs. Similar to P1 TBs, P2 B cells expressed higher levels of surface IgM than others. P2 B cells also expressed increased levels of both CD1c and CD21, when compared to P1 TBs, P3, and P6 B cells. All B cells expressed integrins tested, but both P2 and P4 B

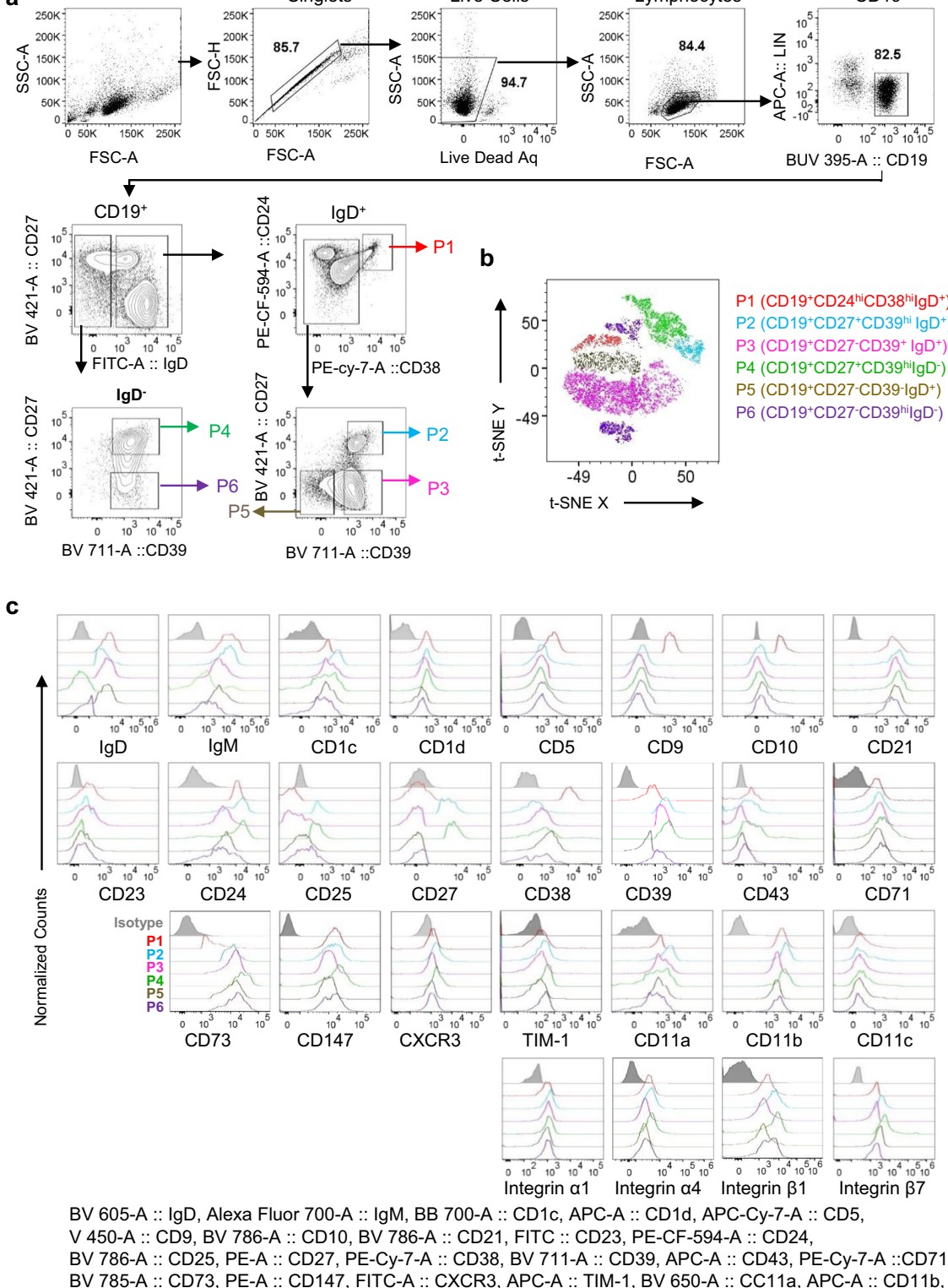

**Fig. 2 Human peripheral blood B cells are composed of six different populations. a** B cells subsets P1 (CD19+CD24hiCD38hiIgD+), P2 (CD19+CD27+CD39hiIgD+), P3 (CD19+CD27−CD39+IgD+), P4 (CD19+CD27+CD39hiIgD−), P5 (CD19+CD27−CD39−IgD+), and P6 (CD19+CD27−CD39+IgD−) B cells in the blood of healthy subjects. Data generated with cells from healthy subjects (n = 5) were combined. **b** A t-SNE plot, showing the distribution of the six B cell populations (P1 to P6). t-SNE plots were made on CD19+ B cells using cell surface markers, including IgD, CD24, CD27, CD38, CD39, with iteration 1000, perplexity 30, and learning rate (eta) 1564. **c** Cell surface expression of IgD, IgM, CD1c, CD1d, CD5, CD9, CD10, CD21, CD23, CD24, CD25, CD27, CD38, CD39, CD43, CCD71, CD73, CD147, CXCR3, TIM-1, CD11a, CD11b, CD11c, integrin α1, integrin α4, integrin β1, and integrin β7 on P1 (red), P2 (light blue), P3 (pink), P4 (light green), P5 (army green), and P6 (violet) B cells. Data generated with cells from healthy subjects (n = 5) were combined.

cells expressed increased levels of some of the cell surface integrins tested (CD11a, CD11b, and integrins α1, α4, β1, and β7; Fig. 2c and Supplementary Fig. 3). Such increased expression of integrins[19] and CXCR3[20] on P2 and P4 B cells suggested that P2 and P4 B cells could be more capable of migrating to target tissues than other B cell populations. Both P2 and P4 B cells also expressed increased levels of CD25 and CD27, but P4, not P2 B cells were IgD− B cells that expressed higher levels of CD71[21], CD73[22], CD147[23], and TIM-1[24] than P2 B cells (Fig. 2c and Supplementary Fig. 2).

Taken together, P1 B cells are CD39[lo/−]CD73[lo] TBs. Both P2 and P4 B cells are from CD24[hi]CD27[+] B10-like B cells. CD39[hi] P2 B cells were further found to be IgD[+]IgM[hi]CD1c[hi]CD21-[hi]CD23[lo/−]CD25[hi] blood circulating marginal zone (MZ)-like B cells. CD39[hi] P4 B cells were found to be IgD[−]IgM[lo]CD1c[+] CD21[+]CD25[hi] memory B cells. In addition, both P2 MZ-like and P4 memory B cells were also different from P1 TBs by expressing increased levels of some of the surface integrins, which might help them migrate to target tissues. They especially P4 memory B cells also expressed increased levels of CD39/ CD73, CD71, CD147, and TIM-1. This also suggests that P4 memory B cells comprise of many of the previously published Breg subsets expressing those molecules[7–10]. We also confirmed CD39 expression on both P2 MZ-like and P4 memory B cells was higher than others before and after activation (Supplementary Fig. 4).

**Both P2 MZ-like and P4 memory B cells are more efficient than P1 TBs and other B cells at suppressing T cell responses**. We next investigated immunosuppressive function of P2 MZ-like and P4 memory B cells, and compared them with those of P1 TBs and other B cell populations. We first examined their ability to express IL-10 by measuring the frequency of IL-10[+] cells (Fig. 3a) and the amount of IL-10 secreted (Fig. 3b) after 48 h stimulation with CpG-B. Both P2 MZ-like and P4 memory B cells were more efficient than P1 TBs and others at expressing IL-10. There was no significant difference between P2 MZ-like and P4 memory B cells. Analysis of IL-10 expression kinetics showed that the frequency of IL-10[+] TBs was higher than those of IL-10[+] P2 MZ-like and P4 memory B cells by 24 h of activation (Fig. 3c). However, this pattern was reversed after 36 h. Nonetheless, the amount of IL-10 secreted from the three B cell subsets were similar by 24 h, but P2 MZ-like and P4 memory B cells secreted more IL-10 than P1 TBs, as measured at 48 h after CpG-B stimulation (Fig. 3d), suggesting that the amount of IL-10 secreted from per cell could be higher by P2 MZ-like and P4 memory B cells than by P1 TBs. All three B cell subsets maintained similar viabilities during in vitro culture (Supplementary Fig. 5), suggesting that the difference in the IL-10 expression kinetics was not related to cell viability. We also measured the amount of IL-6 and TNFα (Supplementary Fig. 6). Consistent with our previous observations[15], P2 MZ-like and P4 memory B cells expressed slightly increased amount of both IL-6 and TNFα, when compared to P1 TBs. However, either IL-6 or TNFα did not significantly affect such B-cell-mediated suppression of Th1 and Th17 response[15]. The roles of PD-L1 expressed on Bregs have been previously reported[15,1,12–14]. In this study, we also found that surface PD-L1 expression was higher on both MZ-like and memory B cells than TBs (Fig. 3e). Summarized data in Fig. 3f further support our conclusion that both P2 MZ-like and P4 memory B cells are more effective than TBs at co-expressing IL-10 and PD-L1. Both MZ-like and memory B cells were also more efficient than TBs at suppressing CD4[+] T cell proliferation (Fig. 3g, h). Experiments using different T:B cell ratios also showed similar results (Fig. 3i).

Although all three B cell subsets were capable of suppressing IFNγ- (Fig. 4a) and IL-17-producing CD4[+] T cell responses (Fig. 4b) elicited by allogeneic DCs (allo-DCs), P2 MZ-like, and P4 memory B cells were also more efficient than P1 TBs. Such decreases of CD4[+] T cell proliferation and cytokine expression were partially recovered by neutralizing IL-10 (Fig. 4c) or PD-L1 activity (Fig. 4d). We thus concluded that P2 MZ-like and P4 memory B cells, originated from B10-like Bregs, were more efficient than P1 TBs at suppressing human CD4[+] T cell responses in vitro.

**P4 memory B cells express *TIGIT* and *GZMB***. We further investigated whether each of these B cell subsets (P1 TBs and P2 MZ-like and P4 memory B cells) could also display additional mechanisms of action to suppress immune responses. We performed a RNA-seq analysis for all six populations (P1–P6) purified from the blood of six healthy individuals. We identified 17278 genes (FDR < 0.05 and genes with sequencing counts per million (CPM) ≥ 1) that were expressed across the different B cell subsets (Supplementary Fig. 7a). Upregulated and downregulated genes were compared between the populations and summarized in Supplementary Fig. 7b. t-SNE clustering analyses of 17,278 genes (left, Fig. 5a) and 1141 genes related to cell surface markers (right, Fig. 5a) show that P1 TBs, as well as P2 MZ-like and P4 memory B cells were relatively well clustered individually. P3 shared a gene profile mostly with P5. In line with the flow cytometry data (Figs. 1 and 2), P2 MZ-like and P4 memory B cells showed a close proximity in the t-SNE clustering of genes for cell surface markers (right, Fig. 5a). List of genes related to cell surface markers used in Fig. 5a (right) are presented in Supplementary Table 1.

Further analysis of the genes upregulated in B10-like Bregs (P2 MZ-like and P4 memory B cells) revealed *TIGIT* (Fig. 5b) that could play an important role in immune regulation[25]. Increased expression of *TIGIT* by P2 MZ-like and P4 memory B cells was observed before (left, Fig. 5b) and after 16 h activation with CpG-B (right, Fig. 5b). Normalized counts for *TIGIT* expression in all six B cell populations are presented in Supplementary Fig. 8a. In addition to *TIGIT*, P2 MZ-like and P4 memory B cells also expressed increased levels of *GZMB* in response to CpG-B (right, Fig. 5b and Supplementary Fig. 8b). *TIGIT* and *GZMB* expression by P1 TBs was minimal. We thus concluded that *TIGIT* and *GZMB* expression by P2 MZ-like and P4 memory B cells further distinguish them from TBs, and other B cell populations tested by expressing increased levels of both *TIGIT* and *GZMB*.

**TIGIT and granzyme B can directly act on T cells to suppress T cell responses**. To further confirm the RNA-seq data, P1–P6 B cells from the blood of healthy individuals were stained with anti-TIGIT and anti-granzyme B antibody. Figure 5c, d shows that P4 memory B cells expressed significantly higher levels of TIGIT and granzyme B than others. P2 MZ-like B cells also expressed increased levels of both TIGIT and granzyme B, when compared to P1 TBs, P3, P5, and P6 B cells. In addition, neutralizing TIGIT with anti-TIGIT antibody resulted in the partial recoveries of IFNγ (upper, Fig. 5e) and IL-17 (lower, Fig. 5e) expression by CD4[+] T cells cocultured with CpG-activated P4 memory B cells. Granzyme B inhibitor (upper, Fig. 5f) also resulted in partial recoveries of both IFNγ and IL-17 expression by CD4[+] T cells cocultured with CpG-activated P1 TBs, P2 MZ-like B cells, and P4 memory B cells. We also found that the changes in IFNγ and IL-17 expression by CD4[+] T cells after inhibiting granzyme B activity were more significant, when CD4[+] T cells were cocultured with P4 memory B cells (P = 0.0013 for IFNγ and P = 0.0011 for IL-17) than P2 MZ-like B cells (P = 0.016 for IFNγ and

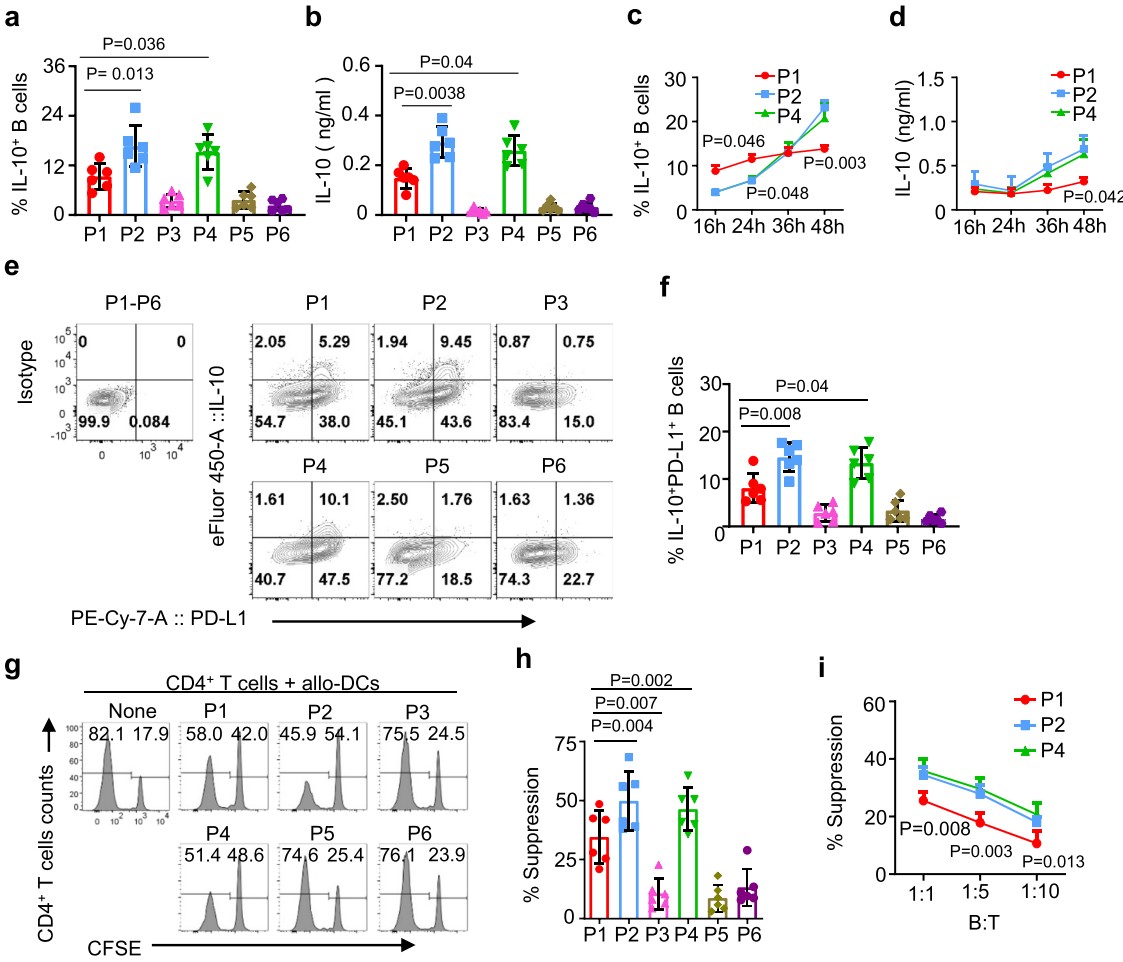

**Fig. 3 P1, P2, and P4 B cells express IL-10 and PD-L1, but P2 and P4 B cells are more efficient than P1 B cells at suppressing T cell responses.**
**a**, **b** Frequency of IL-10+ B cells in P1–P6 B cells (**a**) and the amount of IL-10 secreted by P1, P2, and P4 B cells (**b**). FACS-sorted blood B cells (P1–P6) of healthy subjects ($N = 6$) were incubated for 48 h with 2.5 µg/mL CpG-B before intracellular IL-10 staining. The amount of IL-10 in supernatant was measured by ELISA. Four independent experiments using cells from six healthy subjects (represented by individual dots) were performed. **c**, **d** Kinetics of the frequency of IL-10+ cells (**c**) and the amount of IL-10 secreted (**d**) overtime. Experiments were performed with the same number of cells plated for each indicated populations. Three independent experiments using cells from different healthy subjects ($n = 5$) were performed. **e**, **f** Representative FACS plots (**e**) and summarized data (**f**) for the frequency of IL-10+PD-L1+ B cells. Isotype antibody staining was performed using mixture of P1–P6 B cells. Five independent experiments using cells from different healthy subjects ($n = 6$) were performed. **g–i** Representative FACS data showing the suppression of CD4+ T cell proliferation by P1–P6 B cells (**g**). Summarized data of **g** from five independent experiments using cells from different healthy subjects ($n = 6$) (**h**). CD4+ T cell proliferation assay was performed with indicated numbers of B cells from different healthy subjects ($n = 4$) (**i**). Error bars are mean ± SD. *P* values were determined with one-way ANOVA with Holm–Sidak's multiple comparisons test (**a, b, f,** and **h**) and two-way ANOVA with Dunnett's multiple comparison test (**c, d,** and **i**).

$P = 0.015$ for IL-17) or P1 TBs ($P = 0.028$ for IFNγ and $P = 0.012$ for IL-17). In addition, granzyme B inhibitor did not alter B cell viability (Supplementary Fig. 9).

Although all B cell populations (P1–P6) expressed *TGFB1*, P2 MZ-like and especially P4 memory B cells expressed the highest levels of TGFβ1 than others (Supplementary Fig. 10a, b). Neutralizing TGFβ1 with anti-TGFβ1 antibody also resulted in the enhancement of both IFNγ and IL-17 expression by CD4+ T cells cocultured with CpG-B-activated P4 memory B cells (Supplementary Fig. 10c). Taken together, in addition to their unique cell surface phenotype (including CD39/CD73, CD71, CD147, TIM-1, and several integrins in Fig. 2c), P4 memory B cells further distinguish them from others by expressing increased levels of TIGIT, granzyme B, and TGFβ1 that can further suppress immune responses. We also found that the percentage of TIGIT+ B cells in total CD19+ B cells was higher in the blood than tonsils (Supplementary Fig. 11).

**TIGIT on P4 memory B cells arrests DC maturation and function to control immune response.** TIGIT expressed on Bregs might also act on DCs that express surface CD155 (Fig. 6a). To test this hypothesis, immature monocyte-derived DCs (MDDCs) were stimulated with LPS in the presence of FACS-sorted TIGIT+ memory B cells or TIGIT− B cells (Supplementary Fig. 12). MDDCs alone were also used as controls. After 2 days of coculture, DC maturation was assessed by measuring the expression levels of co-stimulatory molecules (CD80, CD83, CD86, CD40, and ICOS-L) and CCR7. As shown in Fig. 6b, TIGIT+ memory B cells effectively suppressed MDDCs maturation induced by LPS. In addition, MDDCs cocultured with TIGIT+ memory B cells expressed significantly decreased IL-12A and IL-6, when compared to MDDCs cocultured with TIGIT− B cells or MDDCs alone (Fig. 6c). These data (Fig. 6b, c) indicate that TIGIT+ memory B cells are capable of suppressing DC maturation and proinflammatory cytokine expression. Supplementary Figure 13 further

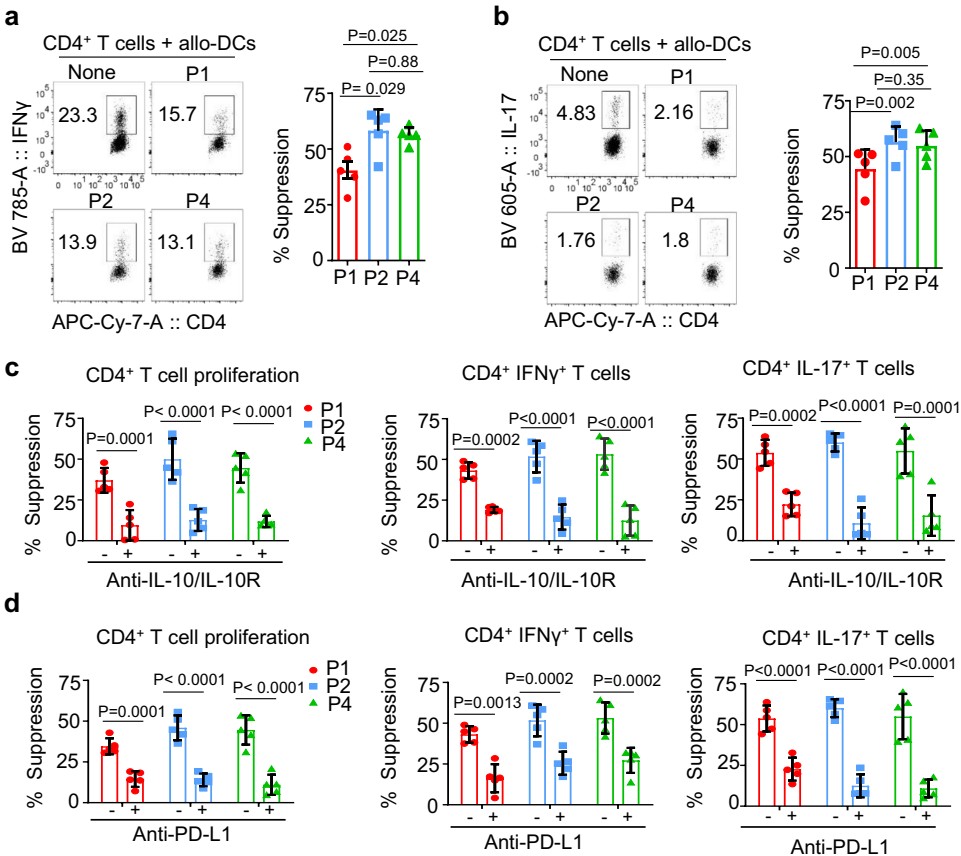

**Fig. 4 P2 and P4 B cells are more efficient than P1 B cells at suppressing allogeneic CD4$^+$ T cell response elicited by allogeneic dendritic cells.** FACS-sorted P1, P2, and P4 B cells from the blood of healthy subjects were stimulated for 48 h with CpG-B and then cocultured for 6 days with CFSE-labeled autologous CD4$^+$ T cells stimulated with allogenic dendritic cells (allo-DCs). CD4$^+$ T cell proliferation was assessed by measuring CFSE dilution. Percent suppression was calculated using the formula as indicated method section. **a**, **b** Representative FACS data (left) and summarized data (right) showing CD4$^+$ IFNγ$^+$ (**a**) and CD4$^+$ IL-17$^+$ (**b**) T cell suppression by FACS-sorted P1, P2, and P4 B cell subsets. Cells were gated based on isotype antibody staining. **c**, **d** Summarized data showing suppression of CD4$^+$ T cells proliferation, as well as IFNγ and IL-17 responses by P1, P2, and P4 B cells in the presence or absence of anti-IL-10/IL-10R (**c**) and anti-PD-L1 (**d**). FACS-sorted P1, P2, and P4 B cells were stimulated for 48 h with CpG-B, preincubated with anti-IL-10/IL-10R or anti-PD-L1 or isotypes, and then cocultured for 6 days with CFSE-labeled CD4$^+$ T cells stimulated with allo-DCs as described in **a** and **b**. Data from five independent experiments performed with cells from different healthy subjects (n = 5). Error bars are mean ± SD. One-way ANOVA with Holm–Sidak's multiple comparisons test (**a**, **b**) and two-way ANOVA with Sidak's multiple comparison test (**c**, **d**) were used.

demonstrated that the inhibition of DC maturation and proinflammatory cytokine expression by TIGIT$^+$ B cells were largely due to the role of TIGIT expressed on memory B cells. Blocking TIGIT with anti-TIGIT neutralizing antibody resulted in the recovery of MDDCs maturation and proinflammatory cytokine expression (Supplementary Fig. 13a, b). This additional mechanism of action displayed by TIGIT$^+$ memory B cells is thought to play a critical role in immune regulation in vivo. DCs are the major antigen-presenting cells that can effectively induce and activate inflammatory responses[26].

To test whether TIGIT$^+$ memory B-cell-modified MDDCs are also capable of controlling T cell response, carboxyfluorescein succinimidyl ester (CFSE)-labeled total CD4$^+$ T cells were cocultured with TIGIT$^+$ B-cell-modified MDDCs for 6 days. As shown in Fig. 6d, TIGIT$^+$ B-cell-modified MDDCs were more efficient at suppressing CD4$^+$ T cell proliferation than the other two groups of MDDCs. MDDCs cocultured with TIGIT$^-$ B cells also suppressed CD4$^+$ T cell proliferation, but it was not statistically significant when compared to MDDCs without coculturing with any B cells. Similarly, IFNγ (Fig. 6e) and IL-17 (Fig. 6f) expression by CD4$^+$ T cells were also significantly suppressed by TIGIT$^+$ Breg-modified MDDCs. Therefore, TIGIT$^+$

memory B cells contribute to immune regulation not only by directly acting on T cells (Fig. 5e), but also by inhibiting DC maturation and proinflammatory functions (Fig. 6).

We next tested whether TIGIT$^+$ memory B cells could also control the induction of CD4$^+$ T cell responses. For this, FACS-sorted naive (CD45RA$^+$CD45RO$^-$CCR7$^+$) CD4$^+$ T cells (Supplementary Fig. 14) were cocultured for 7 days with MDDCs, TIGIT$^+$ B-cell-modified MDDCs, or TIGIT$^-$ B-cell-modified MDDCs, as in Fig. 6. Figure 7a shows that TIGIT$^+$ memory B-cell-modified MDDCs were less efficient than others at inducing CD4$^+$CXCR5$^+$ICOS$^+$ CD4$^+$ T cell responses. In line with this, TIGIT$^+$ memory B-cell-modified MDDCs, compared to MDDCs cocultured with TIGIT$^-$ B cells or MDDCs alone, induced significantly decreased levels of IL-21$^+$ (Fig. 7b) and IL-4$^+$ CD4$^+$ T cell response (Fig. 7c). Interestingly, however, TIGIT$^+$ memory B-cell-modified MDDCs induced significantly enhanced IL-10-producing CD4$^+$ T cell response (Fig. 7d). Taken together, our data indicate that TIGIT$^+$ memory B cells can modify DCs to promote IL-10-producing T cell response, while suppressing ICOS$^+$CXCR5$^+$ CD4$^+$ T cell response along with the inhibition of IL-21 and IL-4 expression. Therefore, TIGIT$^+$ memory B cells might play a critical role in the immune regulation.

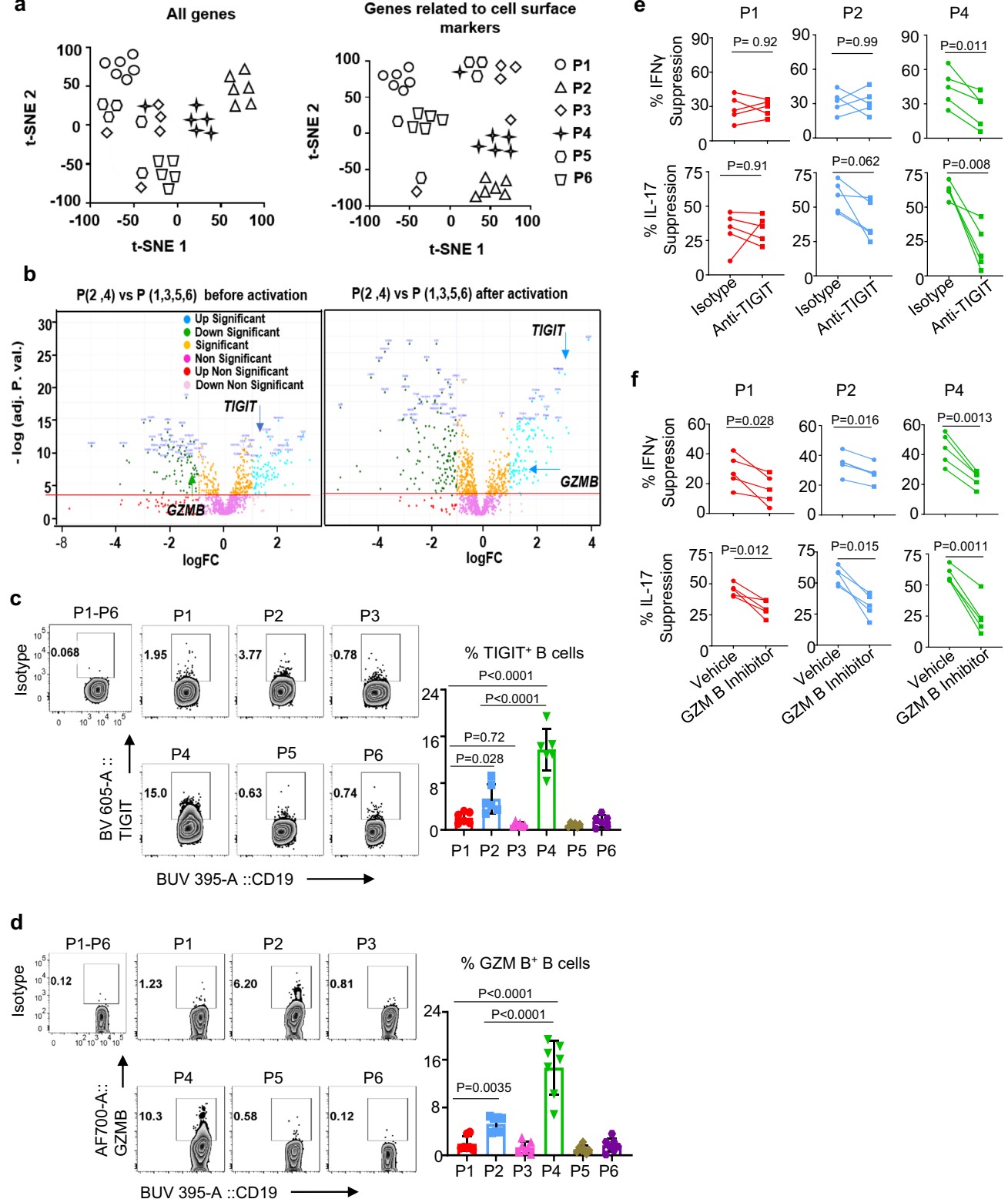

**Bregs in renal and liver allograft recipients.** We next studied Breg subsets, P1 TBs and B10-like Bregs (P2 MZ-like and P4 memory B cells), in the blood of liver (Supplementary Table 2) and kidney (Supplementary Table 3) allograft recipients.

All samples from liver allograft patients were collected at either 2 ($N = 16$) or 5 years ($N = 16$) after transplantation. As shown in Fig. 8a, patients with serum donor-specific antibody (DSA) had decreases in B10-like B cells (Fig. 8a), as well as both P2 MZ-like

(left, Fig. 8b) and P4 memory B cells (right, Fig. 8b). DSA⁻ patients and healthy subjects had similar numbers of those P2 and P4 B cell subsets tested. However, this was not the case of P1 TBs (Supplementary Fig. 15a). Figure 8c further shows that the frequency of $CD25^+CD127^{lo}$ natural naive Tregs[27,28] correlated with the frequency of B10-like (left, Fig. 8c), P2 MZ-like (middle, Fig. 8c), and P4 memory B cells (right, Fig. 8c) expressing TIGIT, but not P1 TBs (Supplementary Fig. 15b). In addition, the

**Fig. 5 P4 memory B cells express functional TIGIT and granzyme B that can directly act on T cells. a** RNA-seq *t*-SNE plots showing clustering of P1–P6 B cells subsets with all genes (left) or genes related to cell surface markers (right). Individual color indicates each of the B cell populations (P1–P6) and each dot represents an individual donor. Data from experiments performed with cells from different healthy subjects ($n = 6$) are presented. **b** RNA-seq volcano plots of differential gene expression between P (2, 4) and P (1, 3, 5, 6) in before (left) and after overnight activation (right) with CpG-B. FC fold change. **c, d** Representative FACS data (left) and summarized data (right) showing the frequency of surface TIGIT (**c**) and intracellular granzyme B (**d**) expression by sort-purified P1–P6 B cells upon 48 h CpG-B stimulation. PMA, ionomycin, brefeldin A, and monensin cocktails were added 5 h before staining them. Following surface staring with CD19 and TIGIT, intracellular granzyme B staining was performed. Data from six independent experiments performed with cells from different healthy subjects ($n = 6$). Isotype antibody staining was performed using mixture of P1–P6 B cells after CpG-B stimulation. Cells were gated based on isotype antibody staining. **e, f** Summarized data showing the suppression of CD4$^+$ T cell responses (IFNγ expression in upper panels and IL-17 expression in lower panels) by P1, P2, and P4 B cells in presence or absence of anti-TIGIT (**e**) and granzyme B blocker (**f**). FACS-sorted P1, P2, and P4 B cells were stimulated for 48 h with CpG-B, preincubated with anti-TIGIT or granzyme B blocker or isotypes, and then cocultured for 4 days with autologous CD4$^+$ T cells stimulated with anti-CD3/anti-CD28 beads. PMA, ionomycin, brefeldin A, and monensin cocktails were added 5 h before staining for intracellular cytokines. Data from four independent experiments performed with cells from different healthy subjects ($n = 5$). Error bars are mean ± SD. *P* values were determined with one-way ANOVA with Holm–Sidak's multiple comparisons test (**c**, **d**) and with a two-tailed paired *t* test (**e**, **f**).

frequency of B10-like B cells (left, Fig. 8d) and P4 memory B cells (right, Fig. 8d) inversely correlated with that of CXCR5$^+$ICOS$^+$ CD4$^+$ T cells. P2 MZ-like B cells also showed a similar trend, but was not statistically significant (middle, Fig. 8d). However, this was not the case for P1 TBs (Supplementary Fig. 15c).

Similar observations were made with blood samples collected at 1 year after renal allograft surgery. The frequency of P4 memory B cells (Fig. 8e) and TIGIT$^+$ memory B cells (Fig. 8f) correlated with that of CD25$^+$CD127$^{lo}$ CD4$^+$ T cells. The frequency of P4 memory B cells (Fig. 8g) and TIGIT$^+$ memory B cells (Fig. 8h) inversely correlated with that of CXCR5$^+$ICOS$^+$ T cells. We also found these were not the case for P1 TBs (Supplementary Fig. 15d, e). Taken together, data in Fig. 8 support that B10-like B cells especially P4 memory B cells expressing surface TIGIT play an important role in the immune regulation in allograft recipients. Data of correlation analyses for healthy subjects are presented in Supplementary Fig. 16a, b. The frequency of TIGIT$^+$ memory B cells, but not B10-like B cells correlated with that of CD25$^+$CD127$^{lo}$ CD4$^+$ T cells (Supplementary Fig. 16a), but it was less significant when compared to that of renal allograft recipients (Fig. 8f). Although there was a trend of an inverse correlation between the frequency of TIGIT$^+$ memory B cells and that of CXCR5$^+$ICOS$^+$ CD4$^+$ T cells, it was not statistically significant (Supplementary Fig. 16b).

## Discussion

This study provides both immunological and clinical evidence for the implication of TIGIT$^+$ memory B cells in immune regulation. TIGIT$^+$ memory B cells, a fraction of B10-like Bregs and the human equivalent of murine B10 cells[15], also possess multiple other characters of Bregs. These include their superior capability, when compared to other Bregs, to express IL-10, TGFβ1, and granzyme B[9,15,29], as well as cell surface PD-L1, CD39, CD73, and TIM-1[7,8,10,30]. All together, our data support that TIGIT$^+$ memory B cells could represent human Bregs that are critical for immune regulation.

TIGIT is an inhibitory receptor that is a member of the poliovirus receptor/nectin family, a subset of the immunoglobulin superfamily[31]. TIGIT expressed on Tregs contribute to their suppressor function by limiting proinflammatory Th1 and Th17 responses[25]. Such inhibition in Th1 and Th17 responses were also observed with TIGIT$^+$ memory B cells, but not TIGIT$^-$ B cells Fig. 5. TIGIT on memory B cells could directly act on CD155 expressed on activated T cells[32,33], resulting in the suppression of T cell response. Furthermore, TIGIT expressed on human memory B cells can also effectively suppress DC maturation via the ligation of CD155 expressed on DCs, resulting in the suppression of their proinflammatory responses. This was further supported by our observations that TIGIT$^+$ memory B cells could

suppress DC expression of IL-12 and IL-6. It was also important to note that TIGIT$^+$ memory B cells efficiently suppress CCR7 expression that is critical for DC migration to lymph nodes to initiate immune responses. This further supports the essential roles of TIGIT$^+$ memory B cells in the suppression of ongoing, as well as initiation of inflammatory responses. DCs are the major antigen-presenting cells which can efficiently initiate and control host immune responses toward either immunity or immune tolerance[26,34–36]. In both renal and liver allograft recipients, we found that the frequency of blood circulating TIGIT$^+$ memory B cells correlated with the frequency of Tregs, while it inversely correlated with that of TFH cells. At this moment, we do not have direct evidence to explain such associations, but our data clearly demonstrated that TIGIT$^+$ B cells could modify DC functions to decrease IL-21- and IL-4-producing T cell response, while promoting IL-10-producing T cell response. A previous study[37] also showed that Bregs especially TBs promoted Treg responses in vitro, but this was largely dependent on IL-10 expressed by TBs. The reduced TFH differentiation by TIGIT$^+$ memory B-cell-modified DCs was in line with the decreased expression of ICOS-L on the DCs (Fig. 6b). ICOS–ICOS-L interaction is required for the induction of BCL6, which is a key transcription factor for TFH development[38]. Therefore, it can be postulated that TIGIT$^+$ memory B cells suppress ICOS-L expression on DCs, resulting in decreased TFH responses followed by decreased serum DSA levels in both renal and liver allograft recipients (Fig. 8). DSA has been associated with both acute and chronic rejection in solid organ allografts, and DSA is one of the major causes of chronic rejection in solid organ transplantation[39,40]. The roles of blood circulating CXCR5$^+$ICOS$^+$ CD4$^+$ T cells in de novo DSA antibody responses has been reported in renal allograft recipients[41]. Another study also reported that PD-L1 expressed on Bregs can also suppress antibody responses[30], but the mechanisms for the Breg PD-L1-mediated suppression of antibody response remain to be fully investigated.

TIGIT$^+$ memory B cells were more effective than CD24$^{hi}$CD38$^{hi}$ TBs at suppressing CD4$^+$ T cell proliferation, as well as Th1 and Th17 responses. This is an important finding because it finally allows us to focus our study on TIGIT$^+$ P4 memory B cells to further define subsets of Bregs and their clinical applications in near future. The advantage of focusing on TIGIT$^+$ memory Bregs over other Breg subsets, as immune regulators, might include their capability to express a variety other inhibitory molecules, including IL-10. IL-10 has long been considered as a master regulator expressed by Bregs. Breg PD-L1-mediated immune suppression has also been reported in recent years[30,42]. In this study, we demonstrated that TIGIT$^+$ Bregs, CD24$^{hi}$CD27$^+$CD39$^{hi}$IgD$^-$IgM$^+$CD1c$^+$ P4 memory B cells, are

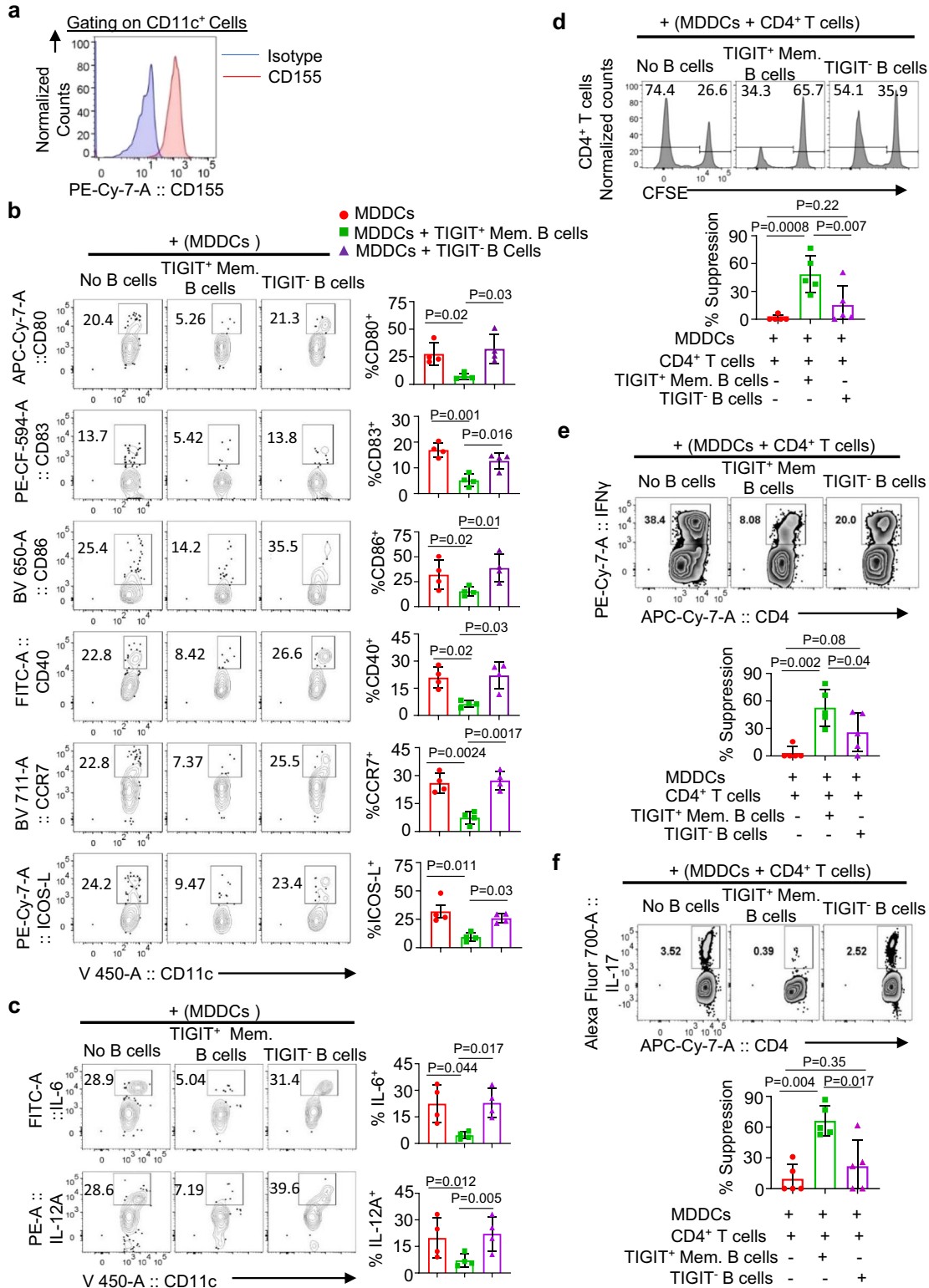

more efficient than CD24hiCD38hi TBs at co-expressing IL-10 and PD-L1. Such increases of PD-L1 expression could be via the IL-10-mediated STAT3 activation[43]. In addition, TIGIT+ P4 memory B cells express higher levels of both TGFβ1 and granzyme B than TBs. B-cell-derived TGFβ1[29] and granzyme B[9] contribute to immune suppression. Phenotype as well as their potent ability to express multiple regulatory effector molecules suggest well-coordinated transcriptional programs that could underpin such unique characteristics of TIGIT+ memory Bregs

that needs to be studied in future. It is also important to note that TIGIT+ human Bregs can also express IL-6 and TNFα. Although, we previously demonstrated that IL-6 and TNFα expressed by human Bregs do not interfere with their ability to suppress IFNγ- and IL-17-producing T cell responses[15], future studies will need to carefully investigate the effects of such cytokines on the Breg-mediated immune regulation.

In addition, TIGIT+ P4 memory B cells are also capable of expressing not only surface CD39/CD73, but also increased levels

**Fig. 6 TIGIT$^+$ memory B cells inhibit LPS-induced dendritic cell activation followed by altered T cell responses. a** Representative FACS plot showing CD155 expression on monocyte-derived dendritic cells (MDDCs). **b, c** Representative FACS plots and summarized data showing the suppression of LPS-induced MDDCs activation (**b**) and proinflammatory cytokines expression (**c**). MDDCs were cultured for 2 days with indicated FACS-sorted B cells (none, TIGIT$^+$ memory B cells, and TIGIT$^-$ B cells) in the presence of 100 ng/mL LPS. Cell surface molecules (**b**) and intracellular IL-6 and IL-12A were stained as indicated. Data from three independent experiments performed with cells from different healthy subjects ($n = 5$). **d–f** FACS plots and summarized data illustrating that TIGIT$^+$ memory B cells suppress MDDC-induced CD4$^+$ T cell proliferation (**d**), as well as IFN$\gamma^+$ (**e**) and IL-17$^+$ (**f**) expression by CD4$^+$ T cells. Purified CD4$^+$ T cells from the blood of healthy individuals were cocultured for 6 days with MDDCs (TIGIT$^+$ memory B-cell-modified MDDCs, TIGIT$^-$ B-cell-modified MDDCs, or MDDCs alone stimulated with 100 ng LPS for 2 days; 5:1 T cells and MDDCs ratio). PMA, ionomycin, brefeldin A, and monensin cocktails were added 5 h before staining them. Data from three independent experiments performed with cells from different healthy individuals ($n = 5$). Cells in **b**, **c**, **e**, and **f** were gated based on isotype control antibody staining. Error bars are mean ± SD. $P$ values were determined with one-way ANOVA with Holm–Sidak's multiple comparisons test (**b–f**).

of TIM-1 that can also contribute to immune regulation[7–10]. CD39$^{hi}$ memory B cells also expressed increased level of CD73. CD39 is a plasma-membrane-bound enzyme that cleaves ATP and ADP down into AMP. AMP is converted into adenosine by CD73 on the cell surface. This sequential activity of the CD39/CD73 pathway scavenges extracellular ATP and generates immunosuppressive adenosine and IL-10[8,44]. Cell surface expression levels of CD39 and CD73 could thus relate to the functional outcomes of P4 memory B cells. In addition, TIGIT$^+$ P4 memory B cells also expressed higher levels of surface integrins (CD11a, CD11b, α1, α4, β1, and β7) than TBs and other B cells, suggesting that effector sites of TIGIT$^+$ Bregs may not always be the same as those of TBs. For example, integrin α4 (CD49d) plays an important role in the recruitment of leukocytes to central nervous system[45] and in the Breg-mediated controlling of experimental autoimmune encephalomyelitis[46]. Integrin β1 and integrin β7 represent the expression of the heterodimer integrin α4β1[47] and α4β7[48,49] leading respectively to homing potential to non-intestinal (α4β7$^-$ and α4β1$^+$) and intestinal tissues (α4β7$^+$). In addition to leukocyte trafficking and migration, integrins (for example, α4) can also contribute to cell activation, survival, and also facilitate interactions between leukocytes and stromal cells found in the marrow or germinal center of lymphoid follicles via VCAM-1 and fibronectin[50], where Bregs might play an important role in controlling antibody responses. CD49d is also known to serve as a signaling receptor that influences B cell survival via the upregulation of Bcl-2 family members[51,52]. It is also known that α1 expressed on tissue resident CD8$^+$ T cells contribute to the increased expression of granzyme B and perforin[53].

In summary, this study demonstrated that TIGIT$^+$ human memory B cells play indispensable roles in immune regulation. Although both TBs and TIGIT$^+$ memory B cells are capable of suppressing T cell responses, TIGIT$^+$ Bregs were more efficient than TBs at suppressing immune responses. This is supported not only by their efficiency to express a variety of immune regulatory molecules, including IL-10, PD-L1, TGFβ1, and granzyme B, as well as surface CD39/73 and TIM-1, but also by suppressing DC maturation and functions via the action of surface TIGIT. Ex vivo data generated with blood samples of two different types of allograft recipients further support the importance of our findings on TIGIT$^+$ memory B cells in immune suppression. Future studies addressing the in vivo functional importance of TIGIT$^+$ memory B cells, as well as TIGIT molecule expressed on subsets of human memory B cells will further help us for the clinical development of TIGIT$^+$ memory Breg therapy. It will also be important to further investigate whether TIGIT$^+$ Bregs represent a distinct and stable lineage, or rather subsets of B cells that can acquire temporally immunosuppressive activity upon certain cues—for example, during the shutdown phases of ongoing immune responses—by investigating transcriptional programmes.

## Methods

**Cell isolation and stimulation.** All experiments using human samples were performed in accordance with the protocols approved by the Institutional Review Board (IRB) in Mayo Clinic and Baylor Scott & White Research Institute. This work was carried out in accordance with the Code of Ethics of the World Medical Association (Declaration of Helsinki). Blood samples from transplant patients were collected with signed consent forms at prespecified time points posttransplant and/or at the time of regular biopsies. Information on renal and liver allograft recipients is summarized in Supplementary Tables 2 and 3. Detection of anti-HLA IgG antibodies was undertaken with LABScreen single-antigen beads (One Lambda) class I (lot 7) and class II (lot 9), according to the manufacturer's protocol. Normalized MFI values are reported. Tonsils from tonsillectomies were also obtained in accordance with IRB approval.

PBMCs and MNCs from tonsils were isolated by density gradient centrifugation using Ficoll-Paque PLUS (GE Healthcare). B cells were purified by negative selection using a pan-B cell enrichment kit (STEMCELL Technologies). Indicated subsets of B cells were sorted with a BD FACSAria II (BD Biosciences). Purified B cells were incubated in the complete RPMI 1640 (cRPMI; Invitrogen) supplemented with 25 mM HEPES (Invitrogen), 1% nonessential amino acids, 2 mM L-glutamate (Invitrogen), 50 µg/mL penicillin, and 50 µg/mL streptomycin (Life Technologies), and 10% fetal bovine serum (FBS) (Gibco). During incubation, cells were stimulated with class B CpG oligodeoxynucleotide (ODN 2006, CpG-B; InvivoGen) at 2.5 µg/mL.

**Measurement of cytokines, granzyme B, PD-L1, and TIGIT expressed on B cells.** A total of $4 \times 10^5$ purified B cells were cultured in cRPMI for indicated time periods in the presence or absence of CpG-B. For intracellular cytokine and granzyme B expression, B cells were further stimulated for 5 h with cell stimulation cocktail containing phorbol PIBM (eBioscience). Human Fc-blocker (Miltenyi Biotec) was added and dead cells were excluded with LIVE/DEAD Fixable Aqua Dead Cell Stain Kit (Life Technologies). Cells were then washed, fixed, and permeabilized using Cytofix/Cytoperm (BD Biosciences), followed by staining them with anti-IL-10 (JES3-9D7), anti-IL-6 (MQ2-13A5), anti-TNFα (MAb11), anti-TGFβ1 (TW4-6H10), or anti-granzyme B antibody (GB11). For surface staining, anti-IgD (IA6-2), anti-CD1c (F10/21A3), anti-CD5 (UCHT2), anti-CD9 (M-L13), anti-CD10 (HI10a), anti-CD21 (B-ly4), anti-CD24 (ML5), anti-CD27 (M-T271) anti-CD38 (HB7), anti-CD39 (Tü66), anti-CD43 (L60), anti-CD11a (Hl111), anti-CD11c (B-ly6), anti-CD49d/integrin α4 (9F10), integrin β7 (FIB504), anti-CD19 (SJ25C1), anti-CD3 (UCHT1), anti-IgM (MHM-88), anti-CD1d (51.1), anti-CD23 (EBVCS-5), anti-CD183/CXCR3 (G025H7), anti-CD11b (CBRM1/5), anti-CD49a/integrin-α1 (TS2/7), anti-CD29/integrin-β1 (TS2/16), anti-TIGIT (VSTM3), anti-CD274/PD-L1 (29E.2A3), and anti-CD365/Tim-1 (1D12) were used. Cells were analyzed with a BD LSR Fortessa flow cytometer using FACSDiva software (v8.0.1; BD Biosciences). Data were analyzed with FlowJo v10 (FlowJo). t-SNE[54] clustering was performed using FlowJo (iteration 1000, perplexity 30, and learning rate (eta) 1564). The amount of IL-10, IL-6, and TNFα secreted by indicated B cells was quantified by bead-based multiplex assays (Millipore Sigma) or ELISA (R&D Systems).

**B cell surface FACS array.** MNCs from tonsils and PBMCs were stimulated for 48 h with CpG-B. PIBM cocktails were added for the last 5 h of the culture. B cells were stained for surface expression of 242 molecules using purified monoclonal antibodies (BD Lyoplate™ Human Cell Surface Marker Screening Panel, BD Biosciences). Alexa Fluor 647-conjugated goat anti-mouse Ig and goat anti-rat Ig secondary antibodies were used to detect the expression of these cell surface markers. Cells were then stained for the measurement of intracellular IL-10 expression. Expression levels of 242 molecules were examined for both intracellular IL-10$^+$ and IL-10$^-$ B cells. First, we acquired ΔMFI values (MFI value of staining antibody subtracted by MFI value of isotype control antibody) for all molecules. Any molecules with ΔMFI ≥ 500 was first selected. Second, we calculated fold changes of ΔMFI values of IL-10$^+$ versus those of IL-10$^-$ B cells. Any molecules showing fold change values ≥ 1.5 was selected to be presented in the heatmap. Color codes in the heatmap was generated with rank of the ΔMFI values of the all

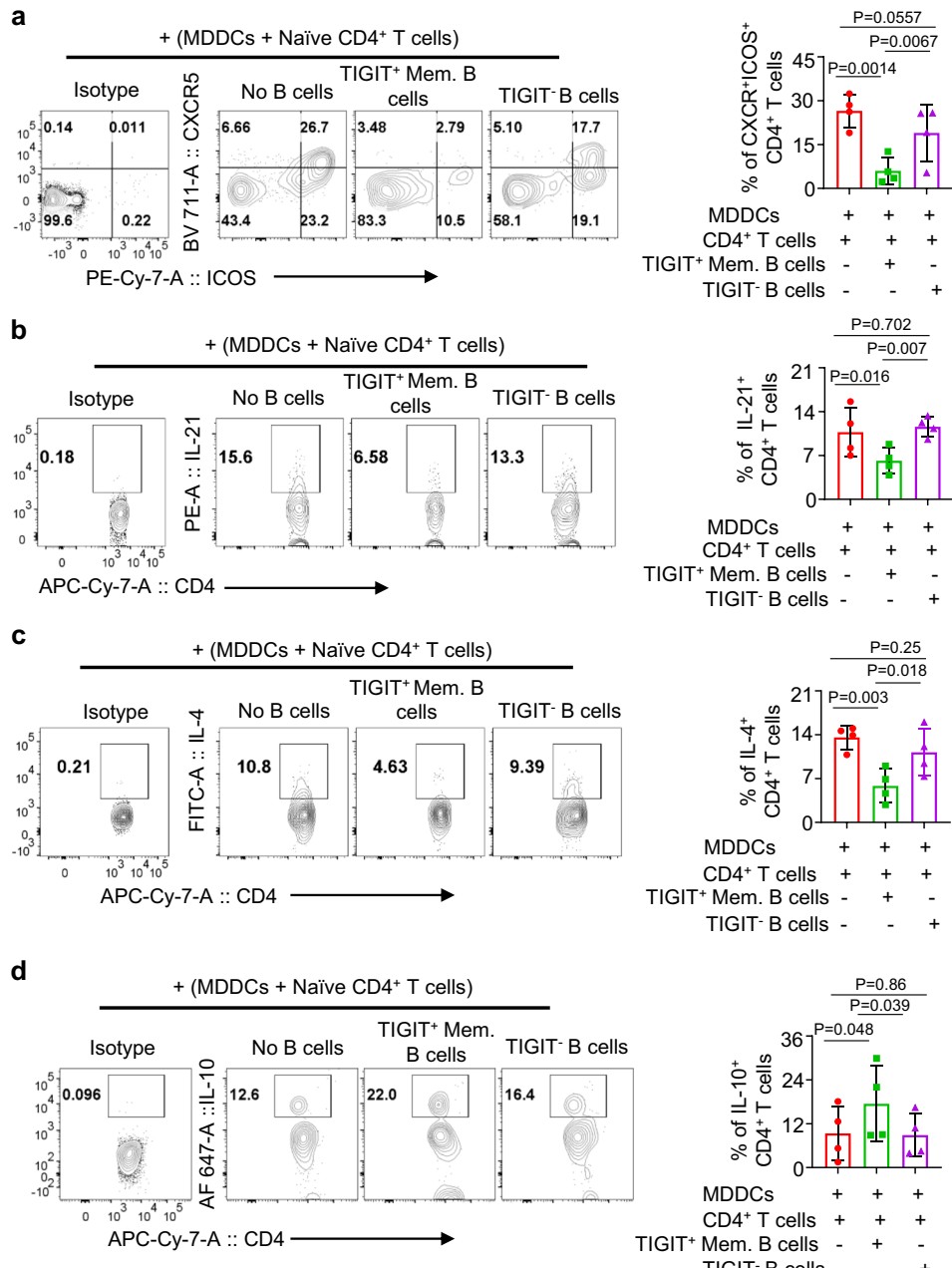

**Fig. 7 TIGIT+ memory B-cell-modified DCs suppress the induction of IL-21- and IL-4-producing T cell response, while promoting the induction of IL-10-producing CD4+ T cell response. a–d** Representative FACS plots (left panels) and summarized data (right panels) showing the induction of CXCR5+ ICOS+ T cell response (**a**), as well as IL-21- (**b**), IL-4- (**c**), and IL-10-expressing (**d**) CD4+ T cell response elicited by monocyte-derived dendritic cells (MDDCs) modified by TIGIT+ memory B cells. TIGIT+ memory B cells or TIGIT− B cells were cocultured with MDDCs for 2 days in the presence of 100 ng/mL LPS before coculturing them for 7 days with FACS-sorted naive CD4+ T cells (CD4+CD45RA+CD45RO−CCR7+). PMA, ionomycin, brefeldin A, and monensin cocktails were added 5 h before intracellular staining of indicated cytokines. Isotype antibody staining was performed using mixture of CD4+ T cells cultured in three different conditions. Cells were gated based on isotype antibody staining. Data from four independent experiments performed with cells from different healthy individuals (n = 4). Error bars are mean ± SD. P values were determined with one-way ANOVA with Holm–Sidak's multiple comparison test (**a–d**).

selected molecules using the Microsoft Excel conditional formatting[55–57], as manufacture's recommendation (https://www.bdbiosciences.com/us/applications/research/stem-cell-research/stem-cell-kits-and-cocktails/human/human-cell-surface-marker-screening-panel/p/560747#tab-0).

**Breg functional assay.** FACS-sorted CD19+CD24hiCD38hiIgD+ (P1), CD19+CD27+CD39hiIgD+ (P2), CD19+CD27−CD39+IgD+ (P3), CD19+CD27+CD39hiIgD− (P4), CD19+CD27−CD39−IgD+ (P5), and CD19+CD27−CD39+IgD− (P6) B cells were stimulated with CpG-B and cocultured for 6 days with CFSE

(Invitrogen)-labeled CD4+ T cells. CD4+ T cells were further stimulated with allo-DCs. Human Fc-blocker was added, and live–dead staining was performed. PIBM was added for the last 5 h. Cell surface was stained with anti-CD4 (RPA-T4), anti-CD19 (SJ25C1), and anti-CD3 (UCHT1). Anti-IFN-γ (B27), anti-TNFα (Mab11), and anti-IL-6 (MQ2-13A5)) were used for intracellular staining. Percent suppression of CD4+ T cell proliferation was calculated by the formula: [(total frequency of proliferating CD4+ T cells cultured with unstimulated B cells or alone − frequency of proliferating CD4+ T cells in the presence of activated B cells)/total frequency of proliferating CD4+ T cells cultured alone or unstimulated B cells] × 100%. Percent suppression of cytokine expression was also calculated in the same way.

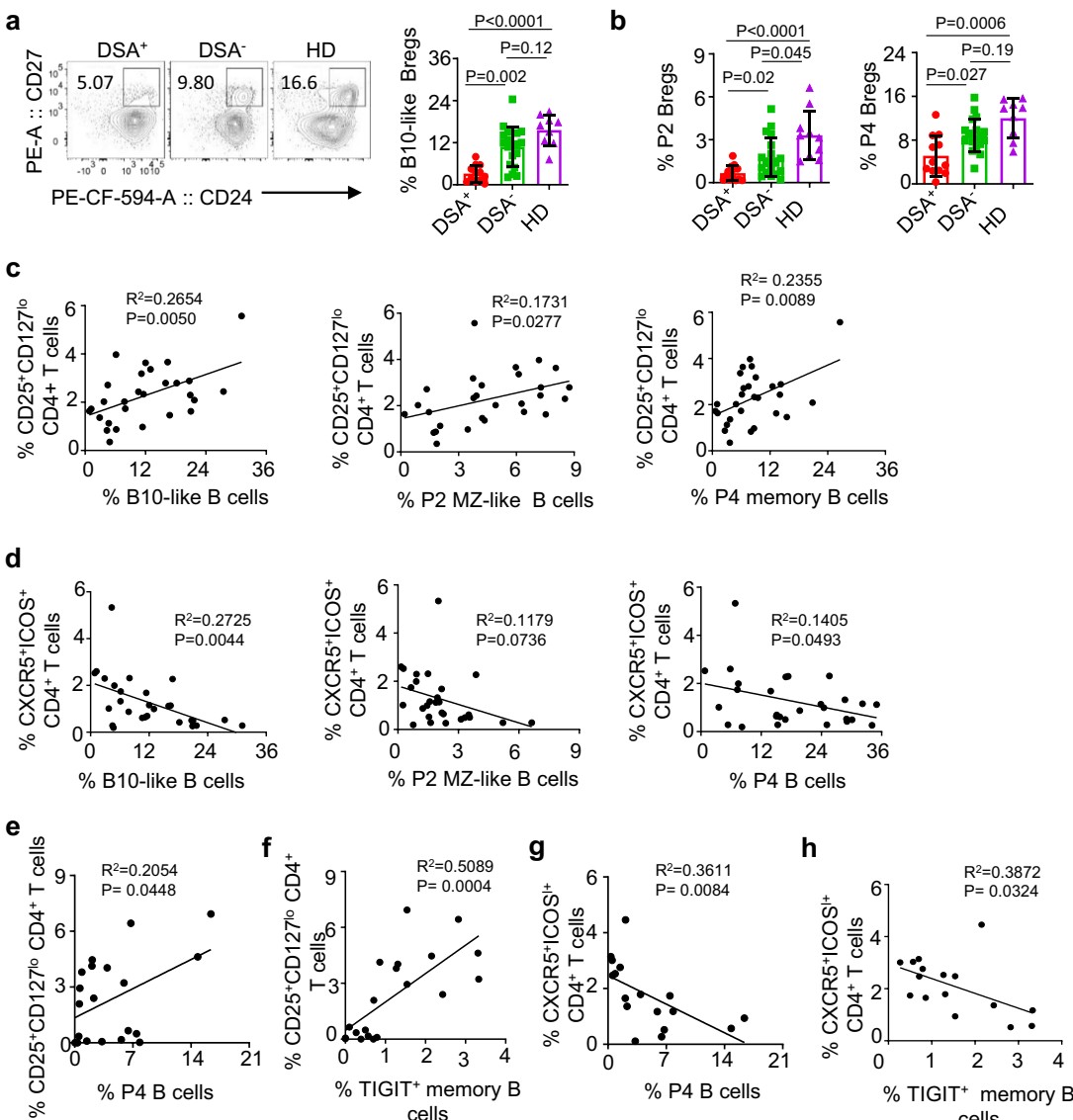

**Fig. 8 TIGIT+ memory B cells in liver and kidney allograft recipients. a**, **b** Representative FACS plots and summarized data for the frequency of CD19+ CD24hiCD27+ B10-like Bregs (**a**), and P2 MZ-like and P4 memory B cells (**b**) in the blood of donor-specific antibody (DSA)+ ($n = 12$) and DSA− ($n = 19$) liver transplant patients and healthy controls ($n = 9$). Dots represent individual patients. **c**, **d** Scatter plots showing the association of the frequency of B10-like, P2 MZ-like, and P4 memory B cells with CD4+CD25+CD127lo Tregs (**c**) and with CXCR5+ICOS+ TFH cells (**d**) in liver allograft recipients. **e**, **f** Scatter plots showing the association of the frequency of P4 memory (**e**) and TIGIT+ memory B cells (**f**) with that of CD4+CD25+CD127lo Tregs in renal allograft recipients. **g**, **h** Scatter plots showing the association of the frequency of P4 memory B cells (**g**) and TIGIT+ memory B cells (**h**) with that of CD4+CXCR5+ ICOS+ TFH in renal allograft recipients. Error bars indicate mean ± SD. $P$ values were determined with one-way ANOVA with Dunnett's multiple comparison test (**a**, **b**). $R^2$ and $P$ values were calculated using Pearson correlation tests (**c**–**h**).

**Blocking experiment**. CpG-B-activated CD19+CD24hiCD38hiIgD+ (P1), CD19+ CD27+CD39hiIgD+ (P2), CD19+CD27−CD39+IgD+ (P3), CD19+CD27+ CD39hiIgD− (P4), CD19+CD27−CD39−IgD+ (P5), and CD19+CD27−CD39+ IgD− (P6) B cells were preincubated with the indicated reagents (anti-IL-10 at 10 µg/mL (JES3-9D7), anti-IL-10R at 10 µg/mL (3F9), and anti-PD-L1 at 10 µg/mL (MIH-1), and anti-TGFβ1 at 10 µg/mL (TW7-28G11), granzyme B blocker at 10 µmol/L (Ac-IEPD-CHO), anti-TIGIT (MBSA43) or control antibodies) for 2 h before coculturing them with others to neutralize targeted molecules. B cells were then cocultured for 6 days with CFSE-labeled CD4+ T cells upon stimulation with allo-DCs. T cell proliferation and cytokine expression were analyzed, as described above.

**Dendritic cell and B cell coculture**. MDDCs were generated by culturing monocytes from healthy donors in cRPMI supplemented with 100 ng/mL GM-CSF (PeproTech) and 50 ng/mL IL-4 (PeproTech)[58]. MDDCs were treated with LPS (100 ng/mL; Invitrogen). LPS before coculturing them with B cells. FACS-sorted TIGIT+ Bregs or TIGIT− B cells were cocultured with MDDCs at 1:1 ratio for 2 days. Cell stimulation cocktail containing PIBM was added 5 h prior to harvest.

Cells were washed, human Fc-blocker was added, and live–dead staining was performed. For surface staining, cells were labeled with anti-CD40 (5C3), anti-CD19 (SJ25C1), anti-CD11c (B-ly6), anti-CD80 (L307.4), anti-CD83(HB15e), anti-CD86 (IT2.2), anti-HLA-DR (L243), anti-CCR7 (150503), and anti-ICOS-L (2D3). Cells were then washed, fixed, and permeabilized using Cytofix/Cytoperm (BD Biosciences), followed by staining them with anti-IL-10 (JES3-9D7), anti-IL-6 (MQ2-13A5), and anti-IL-12A (2Y37).

**Naive CD4+ T cells differentiation assay**. FACS-sorted naive CD4+ T cells were cocultured with or without presence of TIGIT+ Bregs-modified MDDCs (discussed as above) for 6 days at 1:5 TIGIT+ Bregs-modified MDDCs and T cells ratio. Cells were washed, resuspended with 2% FBS containing complete RPMI media, and rested for a day. Cell stimulation cocktail containing PIBM was added 5 h prior to harvest. Cells were washed, human Fc-blocker was added, and live–dead staining was performed. For surface staining, cells were labeled with anti-CD4 (RPA-T4), anti-CD8 (RPA-T8), anti-CD3 (UCHT1), anti-CXCR5 (J252D4), and anti-ICOS. Cells were then washed, fixed, and permeabilized using Cytofix/Cytoperm (BD

Biosciences), followed by staining them with anti-IL-10 (JES3-9D7), anti-IL-4 (MP4-25D2), and anti-IL-21 (3A3-N2.1).

**RNA-seq analysis**. Total RNA was isolated from cell lysates using a modified protocol for the RNAqueous™ Micro Total RNA Isolation Kit (Thermo Fisher Scientific), including on-column DNase digestion. Total RNA was analyzed for quality using the RNA 6000 Pico Kit (Agilent), samples obtained RNA integrity numbers averaging 9.5. NGS library construction was performed using the SMART-Seq v4 Ultra Low Input RNA kit (Clontech) with 5 ng of input total RNA, according to manufacturer's protocol to generate cDNAs. Prior to final library preparation, 15 ng of cDNA was sheared using the Covaris AFA system. The Low Input Library Prep Kit V2 (Clontech) with 2 ng of sheared cDNA was used for final library construction. According to manufacturer's protocol, five PCR cycles were performed for optimal amplification. The quality of the individual libraries was assessed using the High Sensitivity DNA Kit (Agilent). Individual libraries were quantitated using Qubit dsDNA HS Assay Kit (Thermo Fisher Scientific) and equimolar pooled. Final pooled libraries were sequenced on an Illumina NextSeq 500 with paired-end 75 base read lengths. Quality control of raw reads was performed with the FASTQC software (v2.2.3)[59]. Reads were aligned to the human reference genome (GRCh38), using hisat2 (v2.1.0) after quality and adapter trimming by cutadapt (v1.14)[60,61]. Alignment BAM files were processed by samtools (v1.5) and HTSeq-count (v0.9.1) was used to quantify total number of counts for each gene[62,63]. Genes with sequencing CPM < 1 were filtered from further analysis. DESeq2 was used to perform differential gene expression analysis comparing different cell populations at 0 and 16 h (ref. [64]). IPA (QIAGEN Inc., https://www.qiagenbioinformatics.com/products/ingenuitypathway-analysis) was used to perform functional and network analysis for differentially expressed genes. UCSC genome browser was used to visualize the coverage for various genes of interest across cell populations and time points[65]. Highly variable genes across the samples were then calculated and t-SNE clustering were performed using Seurat[66,67]. Function Linnorm.limma () from Linnorm package in R was used for further differential expression analysis in different experimental designs[68]. A volcano plot visualization of DEGs was then plotted, using ggplot function from the R package ggplot2 (ref. [69]). List of genes related to surface markers (Supplementary Table 1) was generated (https://wlab.ethz.ch/cspa/#abstract).

**Statistical analysis**. Statistical analysis was performed with Prism Software (GraphPad) by Student's t test or ANOVA, as appropriate. All data are shown as mean ± SD. A P value <0.05 was considered significant.

**Reporting summary**. Further information on research design is available in the Nature Research Reporting Summary linked to this article.

## Data availability
All sequencing data have been deposited in the National Centre for Biotechnology Information Gene Expression Omnibus (GEO; https://www.ncbi.nlm.nih.gov/geo/). All other relevant data and scripts are available on request without any restrictions.

## Code availability
The accession number for the sequencing data reported in this paper is GEO: GSE151415.

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

## Acknowledgements

We thank nurses and research coordinators in both Mayo Clinic and Baylor Health Care System, who helped patient recruitments and blood sample collections. We also thank the Genomics core in the Baylor Scott & White Research Institute for helping RNA-seq experiment and data analysis. This work was supported by NIH 1 R01 AI 105066 (S.O.), Pirnie Transplantation Award (S.O. and H.J), and Mayo Clinic Foundation (S.O. and H.J.).

## Author contributions

M.M.H., S.S.N., L.T-S., H.J., and S.O. designed experiments. M.M.H. performed experiments. M.M.H., H.J., and S.O. analyzed data. S.S.N., J.G.O., H.J., and S.O. designed patient ex vivo experiments and analyzed data. S.S.N., J.G.O., G.B.K., W.P., M.S., and R.H. identified patients, provided patient samples, and analyzed patient data. M.M.H., V.N., J.W., and S.O. analyzed RNA-seq data. M.M.H., H.J., and S.O. wrote manuscript.

## Competing interests

The authors declare no competing interests.
