## [Peer Review File · Nature Communications]

REVIEWER COMMENTS

Reviewer #1 (Remarks to the Author):

Review of the manuscript entitled 'Critical roles of TIGIT+ human memory Bregs in immune regulation' by Dr. SangKon Oh and colleagues

The critical role of regulatory B cells (Bregs) in immune regulation is undisputed and IL-10 secretion by such cells is one central mechanism of their immunosuppressive or regulative capacity. It should be noted though that there are quite a few other regulatory mechanisms that Bregs have been shown to use, including TGF β , IL-35, PD-L1, IDO, IgG4, CD1d, CD80, CD86 etc. There is still dispute in the field whether Bregs represent a distinct and stable lineage or rather the a subset of B cells that can acquire temporally immunosuppressive activity upon certain cues – for example, during the shut-down phases of ongoing immune responses.

In this current work, the authors aim at addressing these two important questions. By employing a new gating strategy during FACS analyses on B cells isolated from the blood and tonsils of healthy donors the authors identified a memory B cell subpopulation with a CD24^{hi}CD27⁺CD39^{hi}IgD⁻IgM⁺CD1c⁺ TIGIT^{hi} marker phenotype that express IL-10, granzyme B, TGF β 1, and also surface PD-L1, CD39/CD73. These cells display potent in vitro inhibitory activity (mediated by the combined actions of IL-10, granzyme B and TIGIT) directly towards T cells but also via dendritic cells (DCs) which collectively reduced inflammatory effector T cell and T follicular helper (Tfh) responses and fostered generation of regulatory T cells.

To support biological importance of this identified Breg subpopulation, the authors assessed ex vivo responses of Bregs isolated from patients that underwent renal and liver transplantation and noted a reduction of TIGIT+ memory Bregs and Tregs with concurrent increase in Tfh responses and generation of donor-specific antibodies.

The questions the authors try to address (lineage identity, stability and in vivo potency of different Breg populations) are of clear importance particularly towards the goal of using Bregs for therapeutic intervention.

However, major criticism of the work centers around three substantial shortcomings:

1. There are significant concerns with the initial data presentation: All data/experiments in Figures 1 and 2 seem to have been done/based on with scaled data with no statistical analyses applied. Further, all data shown were representative with no 'real' raw data or individual data points shown.
2. The methodological approach and rationale used to identify the 'new' TIGIT+ memory Breg population in the first place: Why was the gating done before the clustering? The current approach introduces a strong bias into the initial data set then utilized to build the subsequent work story. Further, other cells also express CD39/TIGIT etc. albeit in lower amounts. Thus, the population that the authors chose to follow up is not characterized by a novel unique marker profile but rather by the high expression of certain markers (which could represent a temporal or activation-strength induced feature).
3. It is established that CD39 is required for optimal Breg suppressive activity (several publications over the last years) and TIGIT expression on Bregs has recently been identified as a critical to the prevention of tissue inflammation and maintaining CNS tolerance (Xiao, S. et al., Cell reports, 2020). In conjunction with the fact that TIGIT is a known suppressor of T cell effector and Tfh responses, and the lack of true novel mechanistic insights into Breg suppressive activity, the novelty of the observations made here is compromised.
4. The analyses of the patient data are commendable but leave us with correlative data only. There is no true control group in the work that allows to determine what is normal.
5. Thus, the work does not contain a single experiment (ablation of the CD24^{hi}CD27⁺CD39^{hi}IgD⁻IgM⁺CD1c⁺ TIGIT^{hi}) that allows to determine the functional importance (and stability/lineage identity) of these particular cells in vivo.

Therefore, the authors should moderate their language on the importance and potential

therapeutic usage of these cells – particularly in the discussion.

6. The authors focused exclusively on IL-10. However, other cytokines such as IL-6 and LTA are produced by Bregs and are not shown here. Are there any data on those available? This will be an important consideration in therapeutic intervention.

7. The methodological part of the paper lack many details of important assays, for example, how was the isolation of IL-10+ cells done in Figure 1 (capture assay/intracellular staining), the parameters used for t-SNE analysis (i.e. perplexity and iteration values), why exactly were blocking Abs/granzyme B inhibitors added before co-culture, and were they present during the Breg/Tfh/T effector/DC interactions, etc.

Detailed critiques/questions figure by figure:

Figure 1:

1. How exactly were IL-10+ from IL-10- cells isolated? Capture assay? Or was PMA/IO used, which could have large effect on surface molecules.

2. What is the exact percentages of IL-10-expressing cells in (a) – raw data for IL-10 percentages should be shown (e.g. bar graph)

3. How was CD39 detected – are there any statistical analyses of the data at all?

4. The data shown is all scaled – present raw data in supplement.

5. The way the data are presented is confusing. Is the identification of CD39 and the tSNE data done simply by eye (shown color scale?) – where are the statistics to the data?

6. Representation of data as tSNE is unnecessary and difficult to interpret. Raw data should be shown e.g. in density plot and quantified in bar graph, comparing expression levels to CD39- cells.

7. Panel (c): How do CD39-negative cells compare to this analysis?

8. Fig 1a legend: >= instead of <=

Figure 2:

1. CD39 expression is enriched also in MZ.

2. Cells presented in figure 1 were stimulated with CpG (and possibly PMA/IO?). Cells do not appear to be stimulated in this figure, meaning this is inconsistent with the previous figure. Do these markers change with stimulation?

3. Why was gating applied before the tSNE analysis? tSNE should have been done on the raw data.

4. What is the exact rationale for choosing these populations?

5. (b) shows (of course) different clustering because these populations are gating on mutually exclusive markers. This could be detected in an unsupervised manner using tSNE on raw data.

6. Of note, all cells express the integrins, with p2 and p4 having modestly increased amounts

7. Line 147: not P4 memory Bregs... these are just B cells that are CD27+, and should not be called Bregs

Figure 3:

1. There is very little difference in the secretion of cytokines by P1 Secretion over time – could this be due to cell death? Also all populations show similar levels of IL-10 production, which is a bit conspicuous.

2. Were the same number of cells plated for each population?

3. The flow gating can be improved and should be more stringent

4. Where does PD-L1 come from? This marker has not been mentioned before - was PD-L1 not included in the FACS Array, Figure 1?

5. Line 166 'more efficient' at co-expressing? What does this mean? This should maybe be 'more effective'?

6. Panel (g): there is large variation in the depicted data – what statistical analysis was used to obtain this result? As with other figures in the paper, the authors should present all raw data points in their bar graphs

7. Panels (j and k): the effects observed are minimal.

8. It seems as anti-IL10 was not added to the co-cultures? All blocking reagents were applied to the first 12-hours of culture before the suppression assays. What is the rationale behind this?

Figure 4.

1. Memory Bregs express TIGIT and GZMB

2. Can the authors elaborate on the 'stringent filter' they applied? Does this refer to a filter used to generate differentially expressed genes? 0.05 is not particularly stringent?
3. Where are the genes defined related to cell surface markers defined here?
4. Line 186 sp. 'closed each other'
5. Please show isotype staining controls for 4c and d
6. Is GZMB inhibition potentially impacting on B cell viability? – viability data should be shown.
7. On line 208-211, authors perform repeated co-cultures: first with P4, and the same cells then cultured with P2. What is the rationale for this?
8. Overall, the conclusion of this set of data is not convincing: the markers chosen are not unique to the TIGIT+ Bregs, some are rather expressed at slightly higher levels. Also, all other markers are higher as well – which is actually a common feature of CD27+ cells.

Figure 6.

1. Please confirm that indeed an R2 value and not R value is displayed.
2. The data presented are limited to correlative data, but causative analyses/functional experiments utilizing the patient cells are missing.
3. The correlation analysis lacks a control group.
4. The order of the figures is counterintuitive – this would likely be better placed as last figure.

Figure 7.

1. In essence, this figure is a repetition of Figure 5 with another cells type (maybe better in the supplement?).
2. Were all experiments performed with HD cells?
3. CXCR5 and ICOS are activation markers, and likely not representing true Tfh cells. It is hence not unsurprising that Bregs will suppress these.
4. Some flow gating here is unconvincing e.g. IL4 shows no real resolution of a positive population.

Reviewer #2 (Remarks to the Author):

This is a well written and detailed phenotypic and functional analysis of B10-like Bregs based on CD39 and IL-10 expression, among other markers (including PD-L1 and TGF β). The in vitro data are compelling, with suppressive effects noted on proliferation and cytokine expression of T cells, effects on differentiation and cytokine expression by MDDC, and Tfh phenotypes. On the other hand, the in vivo data are largely descriptive and correlative. The extended phenotype of cell surface markers and regulatory effector molecules (including TIGIT) strongly suggest a more coordinated transcriptional programme that underpins the phenotype and function of these "potent suppressor cells". And yet this was not discussed nor interrogated in the RNA-seq dataset. The cell subsets defined would seem to provide a neat experimental framework for probing this in more detail.

Specific comments:

- 1) The authors should state how IL-10 is induced in text at the beginning of the results section.
- 2) It is not clear if the clustering of subsets was defined statistically (ie in an unbiased fashion) or just according to flow phenotypes (fig 2a)? The former would be more robust, and there are well established tools to decipher this.
- 3) What do the authors think is the significance of CD39^{hi} (P2 and P4) and lo (P1 and P5)? Is it relevant functionally, and if so how?
- 4) The RNA-seq data identify TIGIT as a strong candidate to define potent suppressor cell subsets. Granzyme B is also identified. The neutralising experiments seem most convincing for anti-TIGIT than for the GRZB inhibitor, which seemed to exert effects on all populations. What controls were used to test for non-specific inhibitor effects?
- 5) The author should present data describing in depth analysis of the RNA-seq dataset, but with a focus on transcription factors. There are well described candidates that have been reported in regulatory B cells eg PDRM1, AhR, IRF4 etc. This might uncover a signature transcription factor that associates with regulatory function as well as phenotype.
- 6) The in vitro data might be supported further by knockdown of TIGIT to evaluate the effects on regulatory function.

7) In vivo data might be supported further by studying the regulatory potential of TIGIT+ and TIGIT deficient mouse Bregs. This would provide more conclusive evidence for a "critical role for TIGIT" in Breg function.

Minor comments:

Line 151 too - lower case tbs and t

Have the authors examined the distribution of TIGIT in human tonsil?

TIGIT ligand CD155 signals in DCs, how do the authors think TIGIT signals between B cells and T cells?

Reviewer #3 (Remarks to the Author):

The manuscript presents an interesting set of findings, identifying TIGIT as a marker of human Bregs and showing some mechanisms of TIGIT+ Bregs in suppressing effector immune responses. The broad readership of Nature Communications should find this manuscript interesting and sound. The following comments and recommendations focus on improving the clarity of the manuscript and the ability of readers to critically evaluate the findings.

Specific comments:

Line 101: I'm not sure "the differences were less significant" is a statistically meaningful sentence. Some clarity here might be valuable.

Fig 1: It is a bit unclear how the delta MFI is calculated. The methods say delta MFI >500 are shown, but the Fig 1b's scale bar states delta MFI in percentage. Is 100% normalized to the IL-10-cells? It is a little unclear what 0% and 100% mean here. Some clarification would really help readers to interpret and evaluate the results represented by this figure.

Fig 1c: It would help to explain more how the t-SNE was generated and why the authors are drawing the conclusion of "enriched CD39 expression in each population." It seems like what would be more interesting is a t-SNE showing B cell clustering based on surface staining. Then, as an additional dimension of analysis, seeing if an isolated population of cells was CD39+ (or, later TIGIT+). It is also possible this reviewer is misunderstood the data presented. If so, it might be valuable for the authors to put this figure into context either in the results or the methods (or both).

Fig 2b: Is this t-SNE only based on the 4 markers? If so, of course they would segregate into distinct populations. They were gated (thus already defined) as such. It would be more interested to evaluate the clustering with the antibody panel in Fig2c to see if P1, P2, P3, P4, P5, and P6 independently assemble based on this other surface markers. This would essentially be a FACS-based equivalent to Fig 4a (which does the clustering of all genes and then highlights each of the subpopulations on top of the clustering).

Fig 4a: It might be valuable to consider how figures could be interpreted by color-blind individuals (especially more common forms of color-blindness). The red and blue (P1 vs P2) would appear as the same color for many individuals.

Fig 4C-D: Gating is the same for each subset. I'm concerned that each subset might have a different background fluorescence or "stickiness" to the antibodies. It would be helpful to gate based on an isotype control for TIGIT/GZM B for each subpopulation (P1, P2, P3, etc.) individually, if possible. If this was done, it might be helpful to show the isotype control gating for each subpopulation in the supplemental figures.

Fig 6: For instances such as Fig 6b, I'm curious what specific statistics were performed. For multiple comparisons, a one-way ANOVA with a post-test to correct for multiple comparisons should be performed. The methods state "t test or ANOVA" depending on what is appropriate, but it would be helpful to specifically state when each was used.

Fig 6e: I understand P is less than 0.05, but R^2 of 0.2 does not inspire a lot of strength in the correlation. Without 2 of the points in the upper right, this would be a random scatter.

Fig 6: It might be beneficial for the authors to explain why CD25+CD127^{lo} gating was used for the Tregs.

Fig 7: A minor issue, but the "No B cells" has a formatting issue in Fig 7a.

Author Response to Reviewers Comments

Reviewer #1 (Remarks to the Author):

Review of the manuscript entitled 'Critical roles of TIGIT+ human memory Bregs in immune regulation' by Dr. SangKon Oh and colleagues

The critical role of regulatory B cells (Bregs) in immune regulation is undisputed and IL-10 secretion by such cells is one central mechanism of their immunosuppressive or regulative capacity. It should be noted though that there are quite a few other regulatory mechanisms that Bregs have been shown to use, including TGF β , IL-35, PD-L1, IDO, IgG4, CD1d, CD80, CD86 etc. There is still dispute in the field whether Bregs represent a distinct and stable lineage or rather the a subset of B cells that can acquire temporally immunosuppressive activity upon certain cues – for example, during the shut-down phases of ongoing immune responses.

Author response: We thank reviewer's comments. We have described reviewer's points in our manuscript (Lines 43-45, Lines 55-59 and Lines 406-409).

In this current work, the authors aim at addressing these two important questions. By employing a new gating strategy during FACS analyses on B cells isolated from the blood and tonsils of healthy donors the authors identified a memory B cell subpopulation with a CD24^{hi}CD27⁺CD39^{hi}IgD⁻IgM⁺CD1c⁺ TIGIT^{hi} marker phenotype that express IL-10, granzyme B, TGF β 1, and also surface PD-L1, CD39/CD73. These cells display potent in vitro inhibitory activity (mediated by the combined actions of IL-10, granzyme B and TIGIT) directly towards T cells but also via dendritic cells (DCs) which collectively reduced inflammatory effector T cell and T follicular helper (Tfh) responses and fostered generation of regulatory T cells.

To support biological importance of this identified Breg subpopulation, the authors assessed ex vivo responses of Bregs isolated from patients that underwent renal and liver transplantation and noted a reduction of TIGIT+ memory Bregs and Tregs with concurrent increase in Tfh responses and generation of donor-specific antibodies.

The questions the authors try to address (lineage identity, stability and in vivo potency of different Breg populations) are of clear importance particularly towards the goal of using Bregs for therapeutic intervention. However, major criticism of the work centers around three substantial shortcomings:

Author response: We thank reviewer's comments. Our responses to reviewer's comments are summarized below.

1. There are significant concerns with the initial data presentation: All data/experiments in Figures 1 and 2 seem to have been done/based on with scaled data with no statistical analyses applied. Further, all data shown were representative with no 'real' raw data or individual data points shown.

Author response: To present a big set of flow cytometry data (Fig. 1b), we used the heatmap with delta MFI values. In response to reviewer's comment, we present raw delta MFI values of individual markers with statistical analysis in the new Supplementary Fig. 1.

We also replaced t-SNE plots in Fig. 1C with raw delta MFI values of each marker expressed on CD39^{hi} and CD39^{lo/-} B cells with statistical analyses.

Summarized data with individual data points (of Fig. 2C) are presented in new Supplementary Fig. 2 and Supplementary Fig. 3.

2. The methodological approach and rationale used to identify the ‘new’ TIGIT+ memory Breg population in the first place: Why was the gating done before the clustering? The current approach introduces a strong bias into the initial data set then utilized to build the subsequent work story.

Author response: We assume that this comment is related to Figure 2a and 2b. t-SNE analysis in the FlowJo is an unsupervised analysis. FlowJo t-SNE algorithm (based on public data) takes FACS data files (as it is) and generates t-SNE plot in an unsupervised manner. Therefore, our approach does not introduce a bias. For more information, please see this site (<https://docs.flowjo.com/flowjo/advanced-features/dimensionality-reduction/tsne/>) and references cited in the website (Maaten and Hinton 2008 Journal of Machine Learning Research; Wallach and Lilliean 2009 Bioinformatics). The t-SNE analysis in Fig. 2b could also help readers find that 6 populations gated in Fig. 2a are relatively well separated. It also shows the proximity of individual populations as described in Lines 127-131.

Further, other cells also express CD39/TIGIT etc. albeit in lower amounts. Thus, the population that the authors chose to follow up is not characterized by a novel unique marker profile but rather by the high expression of certain markers (which could represent a temporal or activation-strength induced feature).

Author response: As reviewer pointed out, CD39 is not a unique marker for Bregs. By now, however, CD39 (of 242 different cell surface molecules tested) is ubiquitously and most significantly higher on IL-10⁺ B cells than IL-10⁻ B cells that have not been previously reported. New data in Fig. 1c (generated without activating B cells) further show that CD39^{hi} B cells also expressed increased levels of CD71, CD73, and CD147 (than CD39^{lo^{w/-}} B cells) that were reported to be upregulated on subsets of Bregs (described in Lines 47-59). This supports the inclusion of CD39 expression levels in our gating strategy (Fig. 2a) to further subdivide blood B cells into 6 populations, which finally allowed us to determine P4 B cells expressing not only TIGIT, but also other inhibitory molecules, including IL-10, PD-L1, granzyme B and TGFβ1. Indeed, TIGIT expression is largely enriched in P4 followed by P2 (Fig. 4b and 4c). Functional outcomes of TIGIT expression by Bregs (Fig. 4e and 4f, Fig. 5 and Fig. 6) and clinical relevance (Fig. 7) have also been demonstrated. These are important findings that direct us to study more on this B cell subset and also support us to test TIGIT⁺ Bregs in clinic in near future.

At this moment, we do not know if any “novel unique marker of Bregs (as reviewer commented)” is actually existing. Future study will need to address this question by investigating Bregs especially TIGIT⁺ P4 B cells.

3. It is established that CD39 is required for optimal Breg suppressive activity (several publications over the last years) and TIGIT expression on Bregs has recently been identified as a critical to the prevention of tissue inflammation and maintaining CNS tolerance (Xiao, S. et al., Cell reports, 2020). In conjunction with the fact that TIGIT is a known suppressor of T cell effector and Tfh responses, and the lack of true novel mechanistic insights into Breg suppressive activity, the novelty of the observations made here is compromised.

Author response: We agree that CD39 (along with CD73) expressed on B cells can contribute to immune regulation in certain levels. However, it is also important to note that CD39 (or along with CD73 to generate adenosine) is not a universal Breg marker, but it certainly help us further distinguish TIGIT+ memory Bregs from others. For example, CD39 expression is especially high on both MZ-like (P2) and memory Bregs (P4) (as shown in Fig. 2c), but not immature transitional B cells (pop 1 in Fig. 2) that is another major Breg subset in humans, as described in Lines 132-134.

The study mentioned by reviewer mice lacking TIGIT in total B cells. In our study, however, we report that TIGIT expression is mainly enriched on a subset of memory B cells (P4, in humans) that also express high levels of several other key immune regulatory molecules, including IL-10, TGF β 1, granzyme B, CD39/CD73, and TIM-1 which will also play important roles in immune regulation. Our study also demonstrate the mechanisms by which TIGIT⁺ memory Bregs control immune responses by either directly acting on T cells or by controlling DC functions. Therefore, data from this study are novel and also greatly contribute to the advancement of our understanding of the biology of human Bregs and its potential application in clinic in future.

4. The analyses of the patient data are commendable but leave us with correlative data only. There is no true control group in the work that allows to determine what is normal.

Author response: We believe that reviewer missed one of the key messages of the patient data. We demonstrated for the first time that DSA⁺ liver allograft patients have significantly decreased numbers of P2 and P4 memory B cells, but not P1 TBs, when compared to DSA⁻ patients or healthy subjects (Fig. 7a and 7b and Supplementary Fig. 11a). To gain further insight of potential roles of Breg subsets, we analyzed correlations between the frequency of Breg subsets and Tregs or Tfh cells. The frequency of P2 MZ-like and P4 memory B cells correlated with that of Tregs, while inversely correlating with that of Tfh. However, this was not the case for P1 TBs (Supplementary Fig. 11b and c). This study used patient samples collected retrospectively. Due to limited numbers of cells, we were not able to perform any other experiments.

In response to reviewer's comment, we have included correlation data generated with cells from healthy subjects in Supplementary Fig. 14. Text has been revised accordingly (Lines 307-312).

5. Thus, the work does not contain a single experiment (ablation of the CD24hiCD27+CD39hiIgD-IgM+CD1c+ TIGIThi) that allows to determine the functional importance (and stability/lineage identity) of these particular cells in vivo. Therefore, the authors should moderate their language on the importance and potential therapeutic usage of these cells – particularly in the discussion.

Author response: In response to reviewer's suggestion, we removed last sentence in the first paragraph of discussion section – "Data from this study also support a potential clinical development of novel Breg therapy model for certain types of inflammatory diseases where Bregs have been shown to contribute to immune regulation." We have also revised a sentence in the last paragraph of discussion section (Lines 404-406) to be "Although the functional importance of TIGIT⁺ memory Bregs remains to be further elucidated *in vivo*, data from this study could support clinical development of TIGIT⁺ memory Breg therapy in future".

6. The authors focused exclusively on IL-10. However, other cytokines such as IL-6 and LTA are produced by Bregs and are not shown here. Are there any data on those available? This will be an important consideration in therapeutic intervention.

Author response: In response to reviewer's comment, we have included new data showing the amount of IL-6 and TNF α secreted by the three Breg subsets (Supplementary Fig.6). Text has been accordingly revised (Lines 172-176). We have also discussed about these new data in Lines 370-374 (Discussion section) by describing "It is also important to note that TIGIT⁺ human Bregs can also express IL-6 and TNF α . Although we previously demonstrated that IL-6 and TNF α expressed by human Bregs do not interfere with their ability to suppress IFN γ - and IL-17-producing T cell responses (Hasan et al. 2019, reference 15), future studies will need to carefully investigate the effects of such cytokines on the Breg-mediated immune regulation."

7. The methodological part of the paper lack many details of important assays, for example, how was the isolation of IL-10+ cells done in Figure 1 (capture assay/intracellular staining), the parameters used for t-SNE analysis (i.e. perplexity and iteration values), why exactly were blocking Abs/granzyme B inhibitors added before co-culture, and were they present during the Breg/Tfh/T effector/DC interactions, etc.

Author response: As reviewer suggested, we have described the method for detecting IL-10⁺ and IL-10⁻ B cells in the Methods section (Lines 42-49). In response to reviewer #2's suggestion, we have also briefly described this process in Results section (Lines 94-98) to help readers. In Fig. 1a, we have also added the frequency of IL-10⁺ B cells (upper panel) and summary data of multiple experiments using cells from different donors (In response to reviewer's comment in Fig. 1 below). Text and Fig. 1 legend have also been revised accordingly.

As reviewer requested, we have included parameters used for t-SNE analysis in Fig. 2b legend. t-SNE plot (Fig. 2b) was generated with cell surface markers including IgD, CD24, CD27, CD38, CD39, with iteration 1000, perplexity 30, and learning rate (eta) 1564. We have also indicated these in Methods section (Lines 38-39).

We apologize that there was a typo. Blocking reagents were added 2 hrs (not 12 hrs. we have corrected this typo) before co-culturing B cells with others (T and/or DCs) to neutralize targeted molecules (including IL-10, PD-L1, TGF β 1, granzyme B, and TIGIT), as described in Lines 79-80 (Method section). Blocking reagents were in the cultures.

Detailed critiques/questions figure by figure:

Figure 1:

1. How exactly were IL-10+ from IL-10- cells isolated? Capture assay? Or was PMA/IO used, which could have large effect on surface molecules.

Author response: This question has been addressed in reviewer's question #7 above. We understand the reviewer's point that activation could alter expression levels of surface molecules. New data in Fig. 1c show that CD39^{hi} B cells, compared to CD39^{low/-} B cells, also expressed increased levels of CD27, CD71, CD73, and CD147, that are known to be upregulated on Breg subsets (as described in Lines 112-118). B cells in Fig. 1c were stained without activation or any *in vitro* manipulation. Data in supplementary Fig. 4 further show that CD39 expression is higher on P2 and P4 Bregs than others before (Supplementary Fig. 4a) and after activation (Supplementary Fig. 4b).

2. What is the exact percentages of IL-10-expressing cells in (a) – raw data for IL-10 percentages should be shown (e.g. bar graph)

Author response: We have revised Fig. 1a by adding the frequency of IL-10⁺ B cells (upper panel). We have also added new data showing the summary of the frequency of IL-10⁺ B cells in the blood and tonsils (lower panel). Text and Fig. 1 legend have been revised accordingly.

3. How was CD39 detected – are there any statistical analyses of the data at all?

Author response: As shown in Supplementary Fig.1, CD39 expression was ubiquitously and most significantly higher on IL-10⁺ B cells than IL-10⁻ B cells. These are described in Lines 102-104 in this revision. In addition, new Fig. 1c shows that CD39^{hi} B cells, compared to CD39^{low/-} B cells, expressed increased levels of surface molecules (including CD27, CD71, CD73, and CD147) that are known to be expressed on Breg subsets. B cells in Fig. 1c were not activated.

4. The data shown is all scaled — present raw data in supplement.

Author response: As reviewer suggested, we present raw data in Supplementary Fig. 1a (extended data of Fig. 1b) and new Fig. 1c that replace t-SNE plots in Fig. 1c in previous version.

5. The way the data are presented is confusing. Is the identification of CD39 and the tSNE data done simply by eye (shown color scale?) – where are the statistics to the data?

Author response: We have responded to this comment above.

6. Representation of data as tSNE is unnecessary and difficult to interpret. Raw data should be shown e.g. in density plot and quantified in bar graph, comparing expression levels to CD39- cells.

Author response: We have responded to this comment above.

7. Panel (c): How do CD39-negative cells compare to this analysis?

Author response: We have responded to this comment above (Please see new data in Fig. 1c).

8. Fig 1a legend: >= instead of <=

Author response: Thank you. We have corrected it in this revision.

May need to show data showing that CD39 expression on P2 and P4 are higher than others both before and after activation.

Author response: CD39 expression levels on P2 and P4 as well as others (P1, P3, P5, and P6) were presented in Fig. 2c (previous and current versions). B cells in Fig. 2c were not activated. In response to reviewer's comment, however, we present new data in Supplementary Fig. 4, showing that P2 and P4 B cells express higher level of CD39 than others before and after activation. In addition, we also noted that CD39 expression level on P2 and P4 B cells was not dramatically altered after activation with CpG-B. Text has been revised accordingly (Lines 153-155).

Figure 2:

1. CD39 expression is enriched also in MZ.

Author response: As reviewer commented, CD39 expression is enriched also in P2 MZ-like Bregs that are capable of expressing increased levels of IL-10 when compared to other B cells (P1, P3, P5, and P6) as presented in Fig. 2c and new Supplementary Fig. 4.

2. Cells presented in figure 1 were stimulated with CpG (and possibly PMA/IO?). Cells do not appear to be stimulated in this figure, meaning this is inconsistent with the previous figure. Do these markers change with stimulation?

Author response: As reviewer pointed out, cells in Fig. 1a and Fig. 1b were stimulated to capture IL-10⁺ and IL-10⁻ B cells. We found that IL-10⁺ B cells express increased level of CD39 (most significantly as shown in Supplementary Fig. 1). As presented in Supplementary Fig. 4, CD39 expression on P2 and P4 Bregs was not significantly altered by CpG-B stimulation.

In Fig. 1c, we further examined whether CD39 expression could be enriched in human Breg cells by assessing the expression levels of currently known human Breg surface markers on CD39^{hi} and CD39^{/low} B cells. For this, B cells were stained without activation. We found that CD39^{hi} B cells expressed increased levels of CD27, CD71, CD73, and CD147, suggesting its potential value as a human Breg cell surface marker combination with others.

3. Why was gating applied before the tSNE analysis? tSNE should have been done on the raw data.

Author response: As we explained above, t-SNE analysis was done with raw data in an unsupervised manner.

4. What is the exact rationale for choosing these populations?

Author response: Based on our observation in Fig. 1 and Supplementary Fig. 1, we employed surface CD39 expression (relatively well enriched in B cells expressing Breg surface markers) to divide B cells into 6 different populations. FlowJo-supported unsupervised t-SNE analysis also resulted in the 6 populations that we determined with flow gating strategy in Fig. 2a.

5. (b) shows (of course) different clustering because populations are gating on mutually exclusive markers. This could be detected in an unsupervised manner using tSNE on raw data.

Author response: As described above, t-SNE analysis was performed in an unsupervised manner using FlowJo analysis program. All parameters used for unsupervised t-SNE analysis are described in were added in Fig. 2b legend.

6. Of note, all cells express the integrins, with p2 and p4 having modestly increased amounts

Author response: In response to reviewer's point on integrin expression levels, we have revised text to be "All B cells expressed integrins tested, but both P2 and P4 B cells expressed increased levels of some of the cell surface integrins tested (CD11a, CD11b, and integrins α 1, α 4, β 1, and β 7) (Fig. 2c and Supplementary Fig. 3)." in Lines 137-139.

7. Line 147: not P4 memory Bregs... these are just B cells that are CD27+, and should not be called Bregs

Author response: As reviewer suggested, we have called them “P4 memory B cells” throughout this manuscript.

Figure 3:

1. There is very little difference in the secretion of cytokines by P1 Secretion over time — could this be due to cell death? Also all populations show similar levels of IL-10 production, which is a bit conspicuous.

Author response: It is true that there were minor changes in IL-10⁺ cells and the amount of IL-10 secreted by P1 TBs overtime (compared to P2 and P4). As shown in Supplementary Fig. 5, this was not related to cell viability/death, as described in Lines 170-171.

2. Were the same number of cells plated for each population?

Author response: Yes, experiments were performed with the same number of FACS-sorted B cell populations (1×10^5 B cells per well). We have explained this in this revised Figure 3 legend.

3. The flow gating can be improved and should be more stringent

Author response: Gating in Fig. 3e was based on isotype antibody staining. We have included isotype antibody staining data in Fig. 3e. Fig. 3e legend has been revised accordingly.

4. Where does PD-L1 come from? This marker has not been mentioned before - was PD-L1 not included in the FACS Array, Figure 1?

Author response: The roles of PD-L1 expressed on Bregs were previously reported as we described in Line 56 and Line 72. In response to reviewer’s comment, we have re-introduced PD-L1 with references before we present our PD-L1 expression data in Fig. 3a (Line 176). PD-L1 was not in the commercial FACS array kit.

5. Line 166 ‘more efficient’ at co-expressing? What does this mean? This should maybe be ‘more effective’?

Author response: We have revised text (Line 180 in revised version) as reviewer suggested.

6. Panel (g): there is large variation in the depicted data — what statistical analysis was used to obtain this result? As with other figures in the paper, the authors should present all raw data points in their bar graphs

Author response: We assume that reviewer commented on panel (f), not (g). As reviewer suggested, we have revised Fig. 3f by showing individual data points. We used one-way ANOVA with with Holm-Sidak's multiple comparisons test. We have indicated this in Fig.3 legend.

As reviewer recommended, we have also revised Fig. 3J, 3k, 3l, and 3m to show raw data points. We have also presented raw data points in individual bar graphs in Supplementary Figs. 2, 3, 4, 5, 7, 8, 10, 11, 12, and 14,

7. Panels (j and k): the effects observed are minimal.

Author response: B cells especially P2 and P4 B cells were capable of suppressing IFN γ - and IL-17-producing CD4⁺ T cell responses ver 50%, which is significant. We have explained this in revised text (Lines 183-185).

8. It seems as anti-IL10 was not added to the co-cultures? All blocking reagents were applied to the first 12-hours of culture before the suppression assays. What is the rationale behind this?

Author response: We have addressed this question in the main comment #7 above.

Figure 4.

1. Memory Bregs express TIGIT and GZMB

Author response: We assume that reviewer suggests us to write “TIGIT and GZMB” instead of “*TIGIT* and *GZMB*” in the subheading. In that particular section (Lines 190-212), we describe only RNA expression levels of TIGIT and granzyme B. Therefore, we keep “*TIGIT* and *GZMB*” in the subheading. In response to reviewer’s suggestion (comment #7 in Fig. 2 above), we have also call them “P4 memory B cells” not “memory Bregs” in our revised manuscript.

2. Can the authors elaborate on the ‘stringent filter’ they applied? Does this refer to a filter used to generate differentially expressed genes? 0.05 is not particularly stringent?

Author response: We thank reviewer’s comment on this. We should have not used these words “stringent filter”. FDR<0.05 is commonly used for gene expression data analysis. We have revised text accordingly.

3. Where are the genes defined related to cell surface markers defined here?

Author response: All sequencing data in this study have been deposited (GEO: GSE151415). All relevant data and scripts are available on request too. Therefore, we did not present individual genes related to cell surface markers. In response to reviewer’s request, however, we have presented the list of genes related to surface molecules in Supplementary table 1. Text has also been revised accordingly.

In response to reviewer #3’s suggestion, we have also revised Fig. 4a by replacing color codes with different symbols (for color-blinded readers).

4. Line 186 sp. ‘closed each other’

Author response: We have revised this sentence (Lines 199-201 in current version) to be “In line with the flow cytometry data (Fig. 1 and 2), P2 MZ-like and P4 memory B cells showed a close proximity in the t-SNE clustering of genes for cell surface markers (right, Fig. 4a).”

5. Please show isotype staining controls for 4c and d

Author response: We have included isotype antibody staining data in Fig. 4c and 4d in this revision. Fig.4a and 4b legends have also been revised accordingly.

6. Is GZMB inhibition potentially impacting on B cell viability? – viability data should be shown.

Author response: As reviewer requested, we have included cell viability data in Supplementary Fig.9. Granzyme B inhibitor (10 $\mu\text{mol/L}$) did not significantly alter B cell viability. Text has also been revised accordingly (Lines 228-229).

7. On line 208-211, authors perform repeated co-cultures: first with P4, and the same cells then cultured with P2. What is the rationale for this?

Author response: We apologize that the sentence was not clear and make reviewer confused. We have revised text as “We also found that the changes in $\text{IFN}\gamma$ and IL-17 expression by CD4^+ T cells after inhibiting granzyme B activity were more significant when they were co-cultured with P4 memory Bregs ($p < 0.01$) than P2 MZ-like Bregs ($p < 0.05$) or P1 TBs ($p < 0.05$)” in Lines 224-227.

8. Overall, the conclusion of this set of data is not convincing: the markers chosen are not unique to the TIGIT+ Bregs, some are rather expressed at slightly higher levels. Also, all other markers are higher as well — which is actually a common feature of CD27^+ cells.

Author response: We have responded to this question in reviewer’s general comment #2 above. P4 memory B cells can display some common features of memory B cells, but they are a fraction of memory B cells in $\text{CD24}^{\text{hi}}\text{CD27}^+$ human equivalent of murine B cells, as described in our manuscript with references. Although there is no surface marker that is exclusively expressed only on TIGIT + memory Bregs, this study also demonstrates for the first time that P4 memory B cells are the major population that express not only IL-10 and PD-L1, but also TIGIT, granzyme B, and $\text{TGF}\beta 1$ in humans (as described in the conclusion part of this section).

Figure 6.

(As per the reviewer’s request below, previous Fig. 6 is now presented in Fig. 7)

1. Please confirm that indeed an R2 value and not R value is displayed.

Author response: We confirmed that R2 values are presented.

2. The data presented are limited to correlative data, but causative analyses/functional experiments utilizing the patient cells are missing.

Author response: With the number of cells acquired from frozen vials (samples from 1 year, 2 years, or 5 years after transplantation surgery), we were not able to test Breg functions. We will definitely include other important experiments with newly collected patient samples in future.

3. The correlation analysis lacks a control group.

Author response: We have included new data of healthy subjects in Supplementary Fig.14 and described them in text (Lines 310-315).

4. The order of the figures is counterintuitive – this would likely be better placed as last figure.

Author response: As reviewer suggested, patient data are now presented in Fig. 7, last figure. Text has also been revised accordingly.

Figure 7.

(Previous Fig. 7 is now presented in Fig. 6)

1. In essence, this figure is a repetition of Figure 5 with another cells type (maybe better in the supplement?).

Author response: We agree with reviewer point in some degree. However, we also believe that data generated with naïve CD4⁺ T cells have its own messages because it demonstrates that TIGIT⁺ memory B cells can also control the induction of different types of CD4⁺ T cell responses. Therefore, we would like to present this data in Fig. 6. In response to reviewer's comments, however, we removed a separate subheading for the previous Fig. 7 (currently Fig. 6) and rewrite text in Lines 273-285.

2. Were all experiments performed with HD cells?

Author response: All experiments in this Figure were performed with cells from healthy individuals as described in Fig. 6 legend. It will be ideal if we can test patient Bregs, T cells, and DCs in future study (if possible).

3. CXCR5 and ICOS are activation markers, and likely not representing true Tfh cells. It is hence not unsurprising that Bregs will suppress these.

Author response: In response to reviewer's comment, we call them "CXCR5⁺ICOS⁺ T cells" in this revision. This has been corrected Fig. 6 legend and throughout text.

4. Some flow gating here is unconvincing e.g. IL4 shows no real resolution of a positive population.

Author response: Gating was made based on isotype antibody staining that is now presented in Fig. 6. Fig. 6 legend has also been revised accordingly.

Reviewer #2 (Remarks to the Author):

This is a well written and detailed phenotypic and functional analysis of B10-like Bregs based on CD39 and IL-10 expression, among other markers (including PD-L1 and TGFb). The in vitro data are compelling, with suppressive effects noted on proliferation and cytokine expression of T cells, effects on differentiation and cytokine expression by MDDC, and Tfh phenotypes. On the other hand, the in vivo data are largely descriptive and correlative. The extended phenotype of cell surface markers and regulatory effector molecules (including TIGIT) strongly suggest a more coordinated transcriptional programme that underpins the phenotype and function of these "potent suppressor cells". And yet this was not discussed nor interrogated in the RNA-seq dataset. The cell subsets defined would seem to provide a neat experimental framework for probing this in more detail.

Author response: We thank reviewer's comments.

Due to extremely limited numbers of patient cells (in frozen vials), we were not able to do any further experiments. All patient samples used in this study were collected at least one year ago. The number of cells was just enough for performing experiments described in this manuscript. We hope we could get additional data, including more functional data, with prospective sample collection in near future.

In response to reviewer's comment, we have inserted "Phenotype as well as their potent ability to express multiple regulatory effector molecules suggest well-coordinated transcriptional programs that could underpin such unique characteristics of TIGIT+ memory B cells that needs to be studied in future" in Discussion section (Lines 371-373).

Based on the unique characteristics (phenotype and multiple regulatory effector molecules) of TIGIT+ memory B cells, we agree that there could be well-coordinated transcriptional networks/programs that might be controlled by multiple factors. Indeed, we have been addressing this question from the day we acquired RNA-seq data. However, we certainly believe that it will need at least a couple of years from now to have sensible conclusions. It is too early to discuss about transcriptional network in P4 memory Bregs at this moment.

Specific comments:

1) The authors should state how IL-10 is induced in text at the beginning of the results section.

Author response: As reviewer suggested, we have explained how IL-10 was induced and stained in Lines 94-101.

2) It is not clear if the clustering of subsets was defined statistically (ie in an unbiased fashion) or just according to flow phenotypes (fig 2a)? The former would be more robust, and there are well established tools to decipher this.

Author response: t-SNE analysis in Fig. 2b is an unsupervised analysis in that we do not control any parameters in the FACS data files. FlowJo t-SNE algorithm (based on public data) takes FACS data files (as it is) and generates t-SNE plot in an unsupervised manner. Therefore, our approach does not introduce a bias. For more information, please see this site (<https://docs.flowjo.com/flowjo/advanced-features/dimensionality-reduction/tsne/>) and references cited in the website (Maaten and Hinton 2008 Journal of Machine Learning Research; Wallach and Lillien 2009 Bioinformatics).

3) What do the authors think is the significance of CD39hi (P2 and P4) and lo (P1 and P5)? Is it relevant functionally, and if so how?

Author response: CD39 is a plasma-membrane-bound enzyme that cleaves ATP and ADP down into AMP. AMP is converted into adenosine by CD73 on the cell surface. This sequential activity of the CD39/CD73 pathway scavenges extracellular ATP and generates immunosuppressive adenosine and IL-10. Cell surface expression levels of CD39 and CD73 could thus relate to the functional outcomes of Bregs expressing CD39 and CD73. We have described these in Lines 380-385.

4) The RNA-seq data identify TIGIT as a strong candidate to define potent suppressor cell subsets. Granzyme B is also identified. The neutralising experiments seem most convincing for anti-TIGIT than for the GRZB inhibitor, which seemed to exert effects on all populations. What controls were used to test for non-specific inhibitor effects?

Author response: As reviewer pointed out, granzyme B inhibitor exerted effects on all three populations. However, its effect was more significant when T cells were co-cultured with P4 memory B cells ($p < 0.01$) than when they were cultured with P1 or P2 B cells ($p < 0.05$). In the absence of B cells, granzyme B inhibitor alone did not significantly alter T cell responses (not shown). In addition, granzyme B inhibitor did not alter B cell viability (new Supplementary Fig. 9). We have described these in Lines 224-229.

5) The author should present data describing in depth analysis of the RNA-seq dataset, but with a focus on transcription factors. There are well described candidates that have been reported in regulatory B cells eg PDRM1, AhR, IRF4 etc. This might uncover a signature transcription factor that associates with regulatory function as well as phenotype.

Author response: We have responded to this comment above. We agree with the reviewer's point that it is important to uncover TFs that associates with regulatory function and phenotypes. Instead of just providing the list of TFs (shown in the figure, right), we would like to first characterize combined regulatory networks in the Bregs and then report them with their functions. As shown in the figure, multiple TFs are up- and down-regulated regulated in P4 memory B cells compared to others. Indeed, P4 B cells express increased *AHR* (which can contribute to IL-10 expression as previously reported), but they also expressed a sets of TFs that have not been described in regulatory B cells. Therefore, we truly need more time to understand the biology of each of the TBs in P4 memory B cells.

These data have already been deposited in the National Centre for Biotechnology Information Gene Expression Omnibus (GEO). The accession number for the

sequencing data reported in this paper is GEO: GSE151415. Therefore, these data can be assessed by readers at any time.

6) The *in vitro* data might be supported further by knockdown of TIGIT to evaluate the effects on regulatory function.

Author response: All experiments in this study were performed primary human cells. It is thus plausible to employ neutralizing antibodies to demonstrate the regulatory roles of TIGIT expressed on P4 memory Bregs, as we presented in Fig. 4e (TIGIT effect on T cell response) and Supplementary Fig. 12 (TIGIT effect on DC activation).

7) *In vivo* data might be supported further by studying the regulatory potential of TIGIT+ and TIGIT deficient mouse Bregs. This would provide more conclusive evidence for a "critical role for TIGIT" in Breg function.

Author response: We agree with reviewer for the importance of *in vivo* data. TIGIT⁺ Bregs characterized in our study is a fraction of CD27⁺CD24^{hi} human equivalent murine B10-like Bregs. There is still no marker that is known to be specific for this particular B cell subset. Once we discover it in future, we can test it *in vivo* in mice lacking TIGIT expression in that particular B cell subset. Once again, we agree with reviewer on *in vivo* experiment, but we also believe that our data generated in human *in vitro* and patient *ex vivo* are significant.

Minor comments:

Line 151 too - lower case tbs and t

Author response: We have corrected these in Lines 157 in current version of manuscript.

Have the authors examined the distribution of TIGIT in human tonsil?

Author response: TIGIT is mainly enriched on P4 B cells. In response to reviewer's comment, we tested the frequency of TIGIT⁺ CD19⁺ B cells in the blood and tonsils and data are presented in Supplementary Fig. 11. The percentage of TIGIT⁺ B cells was higher in the blood than tonsils tested, as described in Lines 237-239.

TIGIT ligand CD155 signals in DCs, how do the authors think TIGIT signals between B cells and T cells?

Author response: TIGIT on memory B cells could bind to CD155 (carrying ITIM) expressed on activated T cells, resulting in the suppression of T cell response. We have discussed this with references in Lines 331-332.

Reviewer #3 (Remarks to the Author):

The manuscript presents an interesting set of findings, identifying TIGIT as a marker of human Bregs and showing some mechanisms of TIGIT+ Bregs in suppressing effector immune responses. The broad readership of Nature Communications should find this manuscript interesting and sound. The following comments and recommendations focus on improving the clarity of the manuscript and the ability of readers to critically evaluate the findings.

Author response: We thank the reviewer's comments and suggestions to improve our manuscript.

Specific comments:

Line 101: I'm not sure "the differences were less significant" is a statistically meaningful sentence. Some clarity here might be valuable.

Author response: In response to reviewer's suggestion, we have revised text (Lines 113-119) with new Supplementary Fig. 1, showing raw data with statistical analysis.

Fig 1: It is a bit unclear how the delta MFI is calculated. The methods say delta MFI >500 are shown, but the Fig 1b's scale bar states delta MFI in percentage. Is 100% normalized to the IL-10⁻ cells? It is a little unclear what 0% and 100% mean here. Some clarification would really help readers to interpret and evaluate the results represented by this figure.

Author response: We apologize for not providing clear explanation for the methods. To select molecules presented in Fig. 2b, we used two different filters. First, we acquired Δ MFI values (MFI value of staining antibody subtracted by MFI value of isotype control antibody) for all molecules. Any molecules with Δ MFI \geq 500 was first selected. Second, we calculated fold changes of Δ MFI values of IL-10⁺ versus those of IL-10⁻ B cells. Any molecules showing fold change values \geq 1.5 was selected to be presented in the heatmap. Color codes in the heatmap was generated with rank of the Δ MFI values of the all selected molecules using the Microsoft Excel conditional formatting (Xiao et al. 2018 Cancer discovery, Graham G. Walmsley et al. 2015, and Pierre-Edouard Dollet et al. 2016), as manufacture's recommendation (<https://www.bdbiosciences.com/us/applications/research/stem-cell-research/stem-cell-kits-and-cocktails/human/human-cell-surface-marker-screening-panel/p/560747#tab-0>). We have described these processes in Methods section (Lines 49-57). In addition, we have also presented raw data in new Supplementary Fig. 1. We hope this will help readers understand our data better.

Fig 1c: It would help to explain more how the t-SNE was generated and why the authors are drawing the conclusion of "enriched CD39 expression in each population." It seems like what would be more interesting is a t-SNE showing B cell clustering based on surface staining. Then, as an additional dimension of analysis, seeing if an isolated population of cells was CD39⁺ (or, later TIGIT⁺). It is also possible this reviewer is misunderstood the data presented. If so, it might be valuable for the authors to put this figure into context either in the results or the methods (or both).

Author response: As per the request by reviewer 1, we have removed tSNE analysis in Fig. 1c and present raw data of both CD39⁺ and CD39⁻ B cells. Text has also been revised accordingly (Line 112-119).

Fig 2b: Is this t-SNE only based on the 4 markers? If so, of course they would segregate into distinct populations. They were gated (thus already defined) as such. It would be more interested to evaluate the clustering with the antibody panel in Fig2c to see if P1, P2, P3, P4, P5, and P6 independently

assemble based on this other surface markers. This would essentially be a FACS-based equivalent to Fig 4a (which does the clustering of all genes and then highlights each of the subpopulations on top of the clustering).

Author response: t-SNE analysis Fig. 2b was performed on total CD19⁺ B cells with 7 color flow panel including IgD, CD24, CD27, CD38, CD39, CD19 with iteration 1000, perplexity 30, and learning rate (eta) 1564 before gating P1-P6. FlowJo t-SNE algorithm (based on public data) takes FACS data files (as it is) and generates t-SNE plot in an unsupervised manner. For more information, please see this site (<https://docs.flowjo.com/flowjo/advanced-features/dimensionality-reduction/tsne/>) and references cited in the website (Maaten and Hinton 2008 Journal of Machine Learning Research; Wallach and Lillian 2009 Bioinformatics). t-SNE data in Fig. 2b support our gating strategy in Fig. 2a.

Data in Fig. 2c were generated with cells from the same donors in the same experiment, but they (total 27-30 colors) were stained in at least 2-3 different tubes based on our current capacity. Therefore, we were not able to perform t-SNE analysis for all molecules tested in Fig. 2c. In response to reviewer 1's request, we also present new data showing raw MFI values for individual molecules tested in Fig. 2c with statistical analyses (Supplementary Fig. 2 and 3).

Fig 4a: It might be valuable to consider how figures could be interpreted by color-blind individuals (especially more common forms of color-blindness). The red and blue (P1 vs P2) would appear as the same color for many individuals.

Author response: As per reviewer's suggestion, we used different symbols instead of different colors.

Fig 4C-D: Gating is the same for each subset. I'm concerned that each subset might have a different background fluorescence or "stickiness" to the antibodies. It would be helpful to gate based on an isotype control for TIGIT/GZM B for each subpopulation (P1, P2, P3, etc.) individually, if possible. If this was done, it might be helpful to show the isotype control gating for each subpopulation in the supplemental figures.

Author response: We understand reviewer's concern. Before staining, we collected similar numbers of cells from each population in one tube for isotype control antibody staining. Gates are then made based on the isotype staining data (presented in Fig. 4c and 4d), as described in Fig. 4 legend.

Fig 6: For instances such as Fig 6b, I'm curious what specific statistics were performed. For multiple comparisons, a one-way ANOVA with a post-test to correct for multiple comparisons should be performed. The methods state "t test or ANOVA" depending on what is appropriate, but it would be helpful to specifically state when each was used.

Author response: Previous Fig. 6 is now presented in Fig. 7 (as per the reviewer 2's request). We performed one-way ANOVA with Dunn's multiple comparisons test. We revised Fig.7 legend accordingly. We have also revised other Figure legends accordingly.

Fig 6e: I understand P is less than 0.05, but R² of 0.2 does not inspire a lot of strength in the correlation. Without 2 of the points in the upper right, this would be a random scatter.

Author response: We have to agree with the reviewer's point on Fig. 7e (previous Fig. 6e). We believe that this (association between P4 B cells and CD25^{hi}CD127^{low} Tregs) needs to be further investigated

by testing more patient samples. However, the main message in this study still stands with the data in next panels (Fig. 7f and 7h) - TIGIT⁺ memory Bregs show strong association with Tregs as well as with TFH cells.

Fig 6: It might be beneficial for the authors to explain why CD25+CD127^{lo} gating was used for the Tregs.

Author response: In response to reviewer's comment, we have added "natural naïve Tregs" with references in Line 296.

Fig 7: A minor issue, but the "No B cells" has a formatting issue in Fig 7a.

Author response: We have corrected this in Fig. 1 in this revision.

REVIEWERS' COMMENTS

EDITORIAL NOTE: Reviewer #1 was unable to review on this occasion, so Reviewer #3 responded to previous comments on their behalf.

Reviewer #2 (Remarks to the Author):

In their revised manuscript the authors have made a lot of additional changes to the manuscript in response to reviewer comments, largely clarification of technical issues, methodology, changes to figure presentation and annotation, and the addition of raw data, replacing some of the original tSNE plots used in the original manuscript to paired dot plots to describe the B cell phenotypes.

While this more detailed description of the phenotype of IL-10 producing Bregs, together with subset specific comparisons of suppressive functions will be of interest to the community, the study remains largely descriptive and correlative. This reviewer was a little disappointed to read from the rebuttal that confirmatory knock down experiments (eg of TIGIT) were not attempted. Furthermore, the authors provided a heat map in their rebuttal illustrating what appears to be some tantalising data hinting at distinct transcriptional programmes, and yet did not feel in any way obliged to explore these findings further in their revision. Such an analysis, through uncovering of molecular signatures - notably patterns of transcription factors - would be of great value, as distinct from changes in cell surface markers whose expression might be influenced more by culture conditions, mode of stimulation and cell activation or differentiation state. One is left a little uncertain as to whether there exists an unambiguous signature/phenotype representative of immunoregulatory lineages for B cells.

Reviewer #3 (Remarks to the Author):

The manuscript uses an interesting, relatively unbiased, approach to identify TIGIT as a functional marker of human Bregs. They continue to determine the subsequent associated markers of TIGIT+ memory Bregs, and the potency of TIGIT+ Bregs as suppressor of various effector immune responses. The work opens up future questions into the physiological function of this specific B cell subset in humans as well as work into identifying similar markers in murine B cells (allowing for more mechanistic in vivo studies).

The strength of the paper is the in vitro suppressive function of TIGIT+ sorted Bregs, showing this subset is indeed functionally suppressive of APCs and T cells. Despite the in vivo work being largely correlative, these findings are interesting and sound. Furthermore, the reviewer comments are addressed to the satisfaction of this reviewer.

Review of Author Responses to Reviewer #1's comments: This reviewer was requested to address the author's response to Reviewer #1's concerns in lieu of Reviewer #1, as Reviewer #1 was unable to review the revised manuscript. Reviewer #1's primary concerns centered on the lack of statistical rigor and unbiased establishment of the TIGIT+ Breg population. Reviewer #1 also had objections with the clinical reliability of the findings, given a lack of mechanistic data or TIGIT+ Breg ablation. The authors have added raw values for MFI and statistical analyses to accompany summary and representative data. The authors acknowledge the limitations to in vivo data, given the human subject, and have added to the discussion caveats for the clinical relevance of their data.

Reviewer #1 expressed concerns about the methodological details of the in vitro Breg functional suppression assays. The authors responded with additional details critical for the understanding of these in vitro interactions.

Although it is impossible to completely speak for Reviewer #1, it is the opinion of this reviewer that the bulk of Reviewer #1's most pertinent concerns were addressed by the authors.

Author Response to Reviewers Comments

REVIEWERS' COMMENTS

Reviewer #2 (Remarks to the Author):

In their revised manuscript the authors have made a lot of additional changes to the manuscript in response to reviewer comments, largely clarification of technical issues, methodology, changes to figure presentation and annotation, and the addition of raw data, replacing some of the original tSNE plots used in the original manuscript to paired dot plots to describe the B cell phenotypes.

Author response: We thank the reviewer's comments above.

While this more detailed description of the phenotype of IL-10 producing Bregs, together with subset specific comparisons of suppressive functions will be of interest to the community, the study remains largely descriptive and correlative. This reviewer was a little disappointed to read from the rebuttal that confirmatory knock down experiments (eg of TIGIT) were not attempted. Furthermore, the authors provided a heat map in their rebuttal illustrating what appears to be some tantalising data hinting at distinct transcriptional programmes, and yet did not feel in any way obliged to explore these findings further in their revision. Such an analysis, through uncovering of molecular signatures - notably patterns of transcription factors - would be of great value, as distinct from changes in cell surface markers whose expression might be influenced more by culture conditions, mode of stimulation and cell activation or differentiation state. One is left a little uncertain as to whether there exists an unambiguous signature/phenotype representative of immunoregulatory lineages for B cells.

Author response: As we explained previously in the first revision, we confirmed the roles of TIGIT in immune suppression (*in vitro*) using anti-TIGIT antibody that inhibits TIGIT and its ligand interactions. Although an additional *in vitro* experiment (with knocking out TIGIT in memory B cells) could further support our observations, we do not feel that it will significantly improve our manuscript. We believe that an *in vivo* demonstration for the roles of TIGIT expressed on particular subsets of memory B cells will be very important and this work can be done once we further discover TIGIT⁺ memory B cell-specific phenotypes. Regarding the molecular signatures and transcription factors, we presented (to reviewer only) transcription factors that are differentially expressed by different subsets of human B cells. As reviewer mentioned, our observations are tantalizing, indeed. And we have been working on this topic for more than almost 2 years now. We hope we can have our conclusions in a couple of years and publish our data as soon as possible.

We truly appreciate reviewer's comments and suggestions and have acknowledged them in conclusion section of our manuscript (Lines 403-408). In consideration of reviewer's comments, title has also been revised to be "Implication of TIGIT⁺ human memory B cells in immune regulation".

Reviewer #3 (Remarks to the Author):

The manuscript uses an interesting, relatively unbiased, approach to identify TIGIT as a functional marker of human Bregs. They continue to determine the subsequent associated markers of TIGIT+ memory Bregs, and the potency of TIGIT+ Bregs as suppressor of various effector immune responses. The work opens up future questions into the physiological function of this specific B cell subset in humans as well as work into identifying similar markers in murine B cells (allowing for more mechanistic in vivo studies).

Author response: We thank the reviewer's comments above.

The strength of the paper is the in vitro suppressive function of TIGIT+ sorted Bregs, showing this subset is indeed functionally suppressive of APCs and T cells. Despite the in vivo work being largely correlative, these findings are interesting and sound. Furthermore, the reviewer comments are addressed to the satisfaction of this reviewer.

Author response: We thank the reviewer's comments above.

Review of Author Responses to Reviewer #1's comments:

This reviewer was requested to address the author's response to Reviewer #1's concerns in lieu of Reviewer #1, as Reviewer #1 was unable to review the revised manuscript. Reviewer #1's primary concerns centered on the lack of statistical rigor and unbiased establishment of the TIGIT+ Breg population. Reviewer #1 also had objections with the clinical relatability of the findings, given a lack of mechanistic data or TIGIT+ Breg ablation. The authors have added raw values for MFI and statistical analyses to accompany summary and representative data. The authors acknowledge the limitations to in vivo data, given the human subject, and have added to the discussion caveats for the clinical relevance of their data.

Reviewer #1 expressed concerns about the methodological details of the in vitro Breg functional suppression assays. The authors responded with additional details critical for the understanding of these in vitro interactions.

Although it is impossible to completely speak for Reviewer #1, it is the opinion of this reviewer that the bulk of Reviewer #1's most pertinent concerns were addressed by the authors.

Author response: We thank reviewer's comments .